# Adversarially Robust Learning: A Generic Minimax Optimal Learner and Characterization

**Omar Montasser**
TTI-Chicago
omar@ttic.edu

**Steve Hanneke**
Purdue University
steve.hanneke@gmail.com

**Nathan Srebro**
TTI-Chicago
nati@ttic.edu

## Abstract

We present a minimax optimal learner for the problem of learning predictors robust to adversarial examples at test-time. Interestingly, we find that this requires new algorithmic ideas and approaches to adversarially robust learning. In particular, we show, in a strong negative sense, the suboptimality of the robust learner proposed by [21] and a broader family of learners we identify as *local* learners. Our results are enabled by adopting a *global* perspective, specifically, through a key technical contribution: the *global one-inclusion graph*, which may be of independent interest, that generalizes the classical one-inclusion graph due to [17]. Finally, as a byproduct, we identify a dimension characterizing qualitatively and quantitatively what classes of predictors $\mathcal{H}$ are robustly learnable. This resolves an open problem due to [21], and closes a (potentially) infinite gap between the established upper and lower bounds on the sample complexity of adversarially robust learning.

## 1 Introduction

We study the problem of learning predictors that are *robust* to adversarial examples at test-time. Adversarial examples can be thought of as carefully crafted input perturbations that cause predictors to misclassify. Learning predictors robust to adversarial examples is a major contemporary challenge in machine learning. There has been a significant interest lately in how deep learning predictors are *not* robust to adversarial examples [30, 5, 14] – e.g., to adversarial perturbations of bounded $\ell_p$-norms– leading to an ongoing effort to devise methods for learning predictors that *are* adversarially robust.

The aim of this paper is to put forward a theory precisely characterizing the complexity of *robust* learnability. We know from prior work that finite VC dimension is *sufficient* for robust learnability, but we also know that its finiteness is not *necessary* [21]. Furthermore, there is a (potentially) *infinite* gap between the established quantitative upper and lower bounds on the sample complexity of adversarially robust learning [21], and we do not know of any *optimal* learners for this problem. In this paper, we address the following fundamental questions:

*What classes of predictors $\mathcal{H}$ are robustly learnable with respect to an arbitrary perturbation set?*
*Can we design generic optimal learners for adversarially robust learning?*

The problem of characterizing learnability is the most basic question of statistical learning theory. In classical (non-robust) supervised learning, the fundamental theorem of statistical learning [32, 33, 6, 13] provides a complete understanding of *what* is learnable: classes $\mathcal{H}$ with finite VC dimension, and *how* to learn: by the generic learner *empirical risk minimization* ($\mathsf{ERM}_{\mathcal{H}}$). We also know that $\mathsf{ERM}_{\mathcal{H}}$ is a near-optimal learner for $\mathcal{H}$ with sample complexity that is quantified tightly by the VC dimension of $\mathcal{H}$.

**Problem setup.** Given an instance space $\mathcal{X}$ and label space $\mathcal{Y} = \{\pm 1\}$, we consider robustly learning an arbitrary hypothesis class $\mathcal{H} \subseteq \mathcal{Y}^{\mathcal{X}}$ with respect to an arbitrary perturbation set $\mathcal{U}$ :

36th Conference on Neural Information Processing Systems (NeurIPS 2022).

$\mathcal{X} \to 2^{\mathcal{X}}$, where $\mathcal{U}(x) \subseteq \mathcal{X}$ represents the set of perturbations that can be chosen by an adversary at test-time, as measured by the *robust risk*:

$$\mathbb{E}_{(x,y)\sim\mathcal{D}} \left[ \sup_{z\in\mathcal{U}(x)} \mathbb{1}\left\{ h(z) \neq y \right\} \right]. \tag{1}$$

We denote by $\mathrm{RE}(\mathcal{H},\mathcal{U})$ the set of distributions $\mathcal{D}$ over $\mathcal{X} \times \mathcal{Y}$ that are *robustly realizable*: $\exists h^* \in \mathcal{H}, \mathrm{R}_{\mathcal{U}}(h^*;\mathcal{D}) = 0$. A learner $\mathbb{A} : (\mathcal{X} \times \mathcal{Y})^* \to \mathcal{Y}^{\mathcal{X}}$ receives $n$ i.i.d. examples $S = \{(x_i, y_i)\}_{i=1}^n$ drawn from some unknown distribution $\mathcal{D} \in \mathrm{RE}(\mathcal{H},\mathcal{U})$, and outputs a predictor $\mathbb{A}(S)$. The worst-case expected *robust* risk of learner $\mathbb{A}$ with respect to $\mathcal{H}$ and $\mathcal{U}$ is defined as:

$$\mathcal{E}_n(\mathbb{A};\mathcal{H},\mathcal{U}) = \sup_{\mathcal{D}\in\mathrm{RE}(\mathcal{H},\mathcal{U})} \mathbb{E}_{S\sim\mathcal{D}^n} \mathrm{R}_{\mathcal{U}}(\mathbb{A}(S);\mathcal{D}). \tag{2}$$

The *minimax* expected robust risk of learning $\mathcal{H}$ with respect to $\mathcal{U}$ is defined as:

$$\mathcal{E}_n(\mathcal{H},\mathcal{U}) = \inf_{\mathbb{A}} \mathcal{E}_n(\mathbb{A};\mathcal{H},\mathcal{U}). \tag{3}$$

For any $\varepsilon \in (0,1)$, the *sample complexity of realizable robust $\varepsilon$-PAC learning of $\mathcal{H}$ with respect to $\mathcal{U}$*, denoted $\mathcal{M}_{\varepsilon}^{\mathrm{re}}(\mathcal{H},\mathcal{U})$, is defined as

$$\mathcal{M}_{\varepsilon}^{\mathrm{re}}(\mathcal{H},\mathcal{U}) = \min\left\{ n \in \mathbb{N} \cup \{\infty\} : \mathcal{E}_n(\mathcal{H},\mathcal{U}) \leq \varepsilon \right\}. \tag{4}$$

$\mathcal{H}$ is robustly PAC learnable realizably with respect to $\mathcal{U}$ if $\forall_{\epsilon\in(0,1)}$, $\mathcal{M}_{\varepsilon}^{\mathrm{re}}(\mathcal{H},\mathcal{U})$ is finite.

**Related work and gaps.** [21] showed that any class $\mathcal{H}$ with finite VC dimension is robustly PAC learnable with respect to any perturbation set $\mathcal{U}$; by establishing that $\mathcal{M}_{\varepsilon}^{\mathrm{re}}(\mathcal{H},\mathcal{U}) \leq \tilde{O}(\frac{2^{\mathrm{vc}(\mathcal{H})}}{\varepsilon})$, where $\mathrm{vc}(\mathcal{H})$ denotes the VC dimension of $\mathcal{H}$. While this gives a *sufficient* condition for robust learnability, they also showed that finite VC dimension is *not* necessary for robust learnability, indicating a (potentially) infinite gap between the established upper and lower bounds on the sample complexity. We next provide simple motivating examples that highlight these gaps in this existing theory, and suggest that the learner witnessing this upper bound might be very sub-optimal:

**Example 1.** Consider an infinite domain $\mathcal{X}$, the hypothesis class of all possible predictors $\mathcal{H} = \mathcal{Y}^{\mathcal{X}}$, and an all-powerful perturbation set $\mathcal{U}(x) = \mathcal{X}$. In this case, the hypothesis minimizing the population robust risk $\mathrm{R}_{\mathcal{U}}(h;\mathcal{D})$ would always be the all-positive or the all-negative hypothesis, and so these are the only two hypotheses we should compete with. And so, even though $\mathrm{vc}(\mathcal{H}) = \infty$, a single example suffices to inform the learner of whether to produce the all-positive or all-negative function.

**Example 2.** A less extreme and more natural example is to take $\mathcal{X} = \mathbb{R}^\infty$ (an infinite dimensional space), and $\mathcal{H}$ the set of homogeneous halfspaces in $\mathcal{X}$, and a perturbation set $\mathcal{U}(x) = \{z \in \mathcal{X} : \langle x, v \rangle = \langle z, v \rangle$ for $v \in V\}$ where $V$ is the set of the first $d$ standard basis vectors. In this example, an adversary is allowed to *arbitrarily* corrupt all but $d$ features. Note that $\mathrm{vc}(\mathcal{H}) = \infty$ but we can robustly PAC learn $\mathcal{H}$ with $O(d)$ samples: simply project samples from $\mathcal{X}$ onto the subspace spanned by $V$ and learn a $d$-dimensional halfspace.

**Our contributions.** In fact, even more strongly, we show in Theorem 1 that there are problem instances $(\mathcal{H},\mathcal{U})$ that are *not* robustly learnable by the learner proposed by [21], but *are* robustly learnable with a different generic learner. Beyond this, Theorem 1 actually illustrates, in a strong negative sense, the suboptimality of any *local* learner – a family of learners that we identify in this work – which informally *only* has access to labeled training examples and perturbations of the training examples, but otherwise does not know the perturbation set $\mathcal{U}$ (defined formally in Definition 1).

In this work, we adopt a *global* perspective on robust learning. In Section 3, we introduce a novel graph construction, the *global one-inclusion graph*, that in essence embodies the complexity of *robust* learnability. In Theorem 3, for any class $\mathcal{H}$ and perturbation set $\mathcal{U}$, we utilize the global one-inclusion graph to construct a generic *minimax optimal* learner $\mathbb{G}_{\mathcal{H},\mathcal{U}}$ satisfying $\mathcal{E}_{2n-1}(\mathbb{G}_{\mathcal{H},\mathcal{U}};\mathcal{H},\mathcal{U}) \leq 4 \cdot \mathcal{E}_n(\mathcal{H},\mathcal{U})$. Our global one-inclusion graph utilizes the structure of the class $\mathcal{H}$ and the perturbation set $\mathcal{U}$ in a global manner by considering *all* datasets of size $n$ that are robustly realizable, where each dataset corresponds to a vertex in the graph. Edges in the graph correspond to pairs of datasets that agree on $n-1$ datapoints, disagree on the $n^{\mathrm{th}}$ label, and *overlap* on the $n^{\mathrm{th}}$ datapoint according to their $\mathcal{U}$ sets. We arrive at an optimal learner by orienting the edges of this graph to minimize a notion

of *adversarial-out-degree* that corresponds to the average leave-one-out *robust* error. Our learner avoids the lower bound in Theorem 1 since it is *non-local* and utilizes the structure of $\mathcal{U}$ at *test-time*.

In Section 5, we introduce a new complexity measure denoted $\mathfrak{D}_{\mathcal{U}}(\mathcal{H})$ (defined in Equation 12) based on our global one-inclusion graph. We show in Theorem 5 that $\mathfrak{D}_{\mathcal{U}}(\mathcal{H})$ *qualitatively* characterizes robust learnability: a class $\mathcal{H}$ is robustly learnable with respect to $\mathcal{U}$ if and only if $\mathfrak{D}_{\mathcal{U}}(\mathcal{H})$ is finite. In Theorem 6, we show that $\mathfrak{D}_{\mathcal{U}}(\mathcal{H})$ tightly *quantifies* the sample complexity of robust learnability: $\Omega(\frac{\mathfrak{D}_{\mathcal{U}}(\mathcal{H})}{\varepsilon}) \leq \mathcal{M}_{\varepsilon}^{\mathrm{re}}(\mathcal{H}, \mathcal{U}) \leq \tilde{O}(\frac{\mathfrak{D}_{\mathcal{U}}(\mathcal{H})}{\varepsilon})$. This closes the (potentially) *infinite* gap previously established [21].

In Section 6, beyond the realizable setting, we show in Theorem 9 that our complexity measure $\mathfrak{D}_{\mathcal{U}}(\mathcal{H})$ bounds the sample complexity of *agnostic* robust learning: $\mathcal{M}_{\varepsilon}^{\mathrm{ag}}(\mathcal{H}, \mathcal{U}) \leq \tilde{O}(\frac{\mathfrak{D}_{\mathcal{U}}(\mathcal{H})}{\varepsilon^2})$. This shows that $\mathfrak{D}_{\mathcal{U}}(\mathcal{H})$ tightly (up to log factors) characterizes the sample complexity of *agnostic* robust learning, since by definition, $\mathcal{M}_{\varepsilon}^{\mathrm{ag}}(\mathcal{H}, \mathcal{U}) \geq \mathcal{M}_{\varepsilon}^{\mathrm{re}}(\mathcal{H}, \mathcal{U})$.

## 2 Local learners are suboptimal

In this section, we identify a broad family of learners, which we term *local* learners, and show that such learners are *suboptimal* for adversarially robust learning. Informally, *local* learners *only* have access to labeled training examples and perturbations of the training examples, but otherwise do not know the perturbation set $\mathcal{U}$. More formally,

**Definition 1** (Local Learners). For any class $\mathcal{H}$, a local learner $\mathbb{A}_{\mathcal{H}} : (\mathcal{X} \times \mathcal{Y} \times 2^{\mathcal{X}})^* \to \mathcal{Y}^{\mathcal{X}}$ for $\mathcal{H}$ takes as input a sequence $S_{\mathcal{U}} = \{(x_i, y_i, \mathcal{U}(x_i)\}_{i=1}^m \in \mathcal{X} \times \mathcal{Y} \times 2^{\mathcal{X}}$ consisting of labeled training examples and their corresponding perturbations according to some perturbation set $\mathcal{U}$, and outputs a predictor $f \in \mathcal{Y}^{\mathcal{X}}$. In other words, $\mathbb{A}$ has full knowledge of $\mathcal{H}$, but only *local* knowledge of $\mathcal{U}$ through the training examples.

We note that the robust learner proposed by [21], for example, *is* a local learner: for a given a class $\mathcal{H}$ and input $S_{\mathcal{U}} = \{(x_i, y_i, \mathcal{U}(x_i)\}_{i=1}^m$, their learner outputs a majority-vote over predictors in $\mathcal{H}$, that are carefully chosen based on the input $S_{\mathcal{U}}$. Moreover, adversarial training methods in practice [e.g., 20, 35] are also examples of local learners (they only utilize the perturbations on the training examples). We provably show next that *local* learners are *not* optimal. We give a construction where it is not possible to robustly learn without taking advantage of the information about $\mathcal{U}$ at *test-time*.

**Theorem 1.** *There is an instance space $\mathcal{X}$ and a class $\mathcal{H}$, such that for any* local *learner $\mathbb{A}_{\mathcal{H}} : (\mathcal{X} \times \mathcal{Y} \times 2^{\mathcal{X}})^* \to \mathcal{Y}^{\mathcal{X}}$ and any sample size $m \in \mathbb{N}$, there exists a perturbation set $\mathcal{U}$ for which:*

1. *$\mathbb{A}_{\mathcal{H}}$ fails to robustly learn $\mathcal{H}$ with respect to $\mathcal{U}$ using $m$ samples.*

2. *There exists a non-local learner $\mathbb{G}_{\mathcal{H}, \mathcal{U}} : (\mathcal{X} \times \mathcal{Y})^* \to \mathcal{Y}^{\mathcal{X}}$ which robustly learns $\mathcal{H}$ with respect to $\mathcal{U}$ with $0$ samples.*

This negative result highlights that there are limitations to what can be achieved with *local* learners. It also highlights the importance of utilizing the structure of the perturbation set $\mathcal{U}$ at test-time, which has been observed in the context of *transductive* robust learning where the learner receives a training set of $n$ labeled examples and a test set of $n$ unlabeled adversarial perturbations, and is asked to label the test set with few errors [23]. In practice, randomized smoothing [8] is an example of a non-local method in the sense that at prediction time, it uses the perturbation set to compute predictions.

*Proof of Theorem 1.* We begin with describing the instance space $\mathcal{X}$ and the class $\mathcal{H}$. Pick three infinite unique sequences $(x_n^+)_{n \in \mathbb{N}}, (x_n^-)_{n \in \mathbb{N}}$, and $(z_n)_{n \in \mathbb{N}}$ from $\mathbb{R}^2$ such that for each $n \in \mathbb{N} : x_n^+ = (n, 1), x_n^- = (n, -1), z_n = (n, 0)$, and let $\mathcal{X} = \cup_{n \in \mathbb{N}} \{x_n^+, x_n^-, z_n\}$. Consider the class $\mathcal{H}$ defined by

$$\mathcal{H} = \left\{ h_{\boldsymbol{y}} : \boldsymbol{y} \in \{\pm 1\}^{\mathbb{N}} \right\}, \text{ where } h_{\boldsymbol{y}}(z_n) = y_n \wedge h_{\boldsymbol{y}}(x_n^+) = +1 \wedge h_{\boldsymbol{y}}(x_n^-) = -1 \, (\forall n \in \mathbb{N}) . \quad (5)$$

Observe that all classifiers in $\mathcal{H}$ are constant on $(x_n^+)_{n \in \mathbb{N}}$ and $(x_n^-)_{n \in \mathbb{N}}$, but they shatter $(z_n)_{n \in \mathbb{N}}$. We will consider a *random* perturbation set $\mathcal{U} : \mathcal{X} \to 2^{\mathcal{X}}$ that is defined as follows:

$$\forall n \in \mathbb{N} : \begin{cases} \mathcal{U}(x_n^+) = \{x_n^+, z_n\} \text{ and } \mathcal{U}(x_n^-) = \{x_n^-\} \text{ and } \mathcal{U}(z_n) = \{x_n^+, x_n^-, z_n\} \text{ w.p. } \frac{1}{2}, \\ \mathcal{U}(x_n^+) = \{x_n^+\} \text{ and } \mathcal{U}(x_n^-) = \{x_n^-, z_n\} \text{ and } \mathcal{U}(z_n) = \{x_n^+, x_n^-, z_n\} \text{ w.p. } \frac{1}{2}. \end{cases} \quad (6)$$

For any sample size $m \in \mathbb{N}$, let $P$ be a uniform distribution on
$$\left\{(x_1^+, +1), (x_1^-, -1), \ldots, (x_{3m}^+, +1), (x_{3m}^-, -1)\right\}.$$
Observe that for any randomized $\mathcal{U}$ (according to Equation 6), the distribution $P$ is *robustly realizable* with respect to $\mathcal{U}$: $\exists h \in \mathcal{H}, \mathrm{R}_{\mathcal{U}}(h; P) = 0$. Let $\mathbb{A}$ be an arbitrary *local* learner (see Definition 1), i.e., $\mathbb{A}$ has full knowledge of the class $\mathcal{H}$, but only partial knowledge of $\mathcal{U}$ through the training samples. Let $S \sim P^m$ be a fixed random set of training examples drawn from $P$. Then,

$$
\begin{aligned}
\mathop{\mathbb{E}}_{\mathcal{U}} \mathrm{R}_{\mathcal{U}}(\mathbb{A}(S_{\mathcal{U}}); P) &= \mathop{\mathbb{E}}_{\mathcal{U}} \mathop{\mathbb{E}}_{(x,y)\sim P} \mathbb{1}[\exists z \in \mathcal{U}(x) : \mathbb{A}(S_{\mathcal{U}})(z) \neq y] \\
&\geq \mathop{\Pr}_{(x,y)\sim P}[(x,y) \notin S] \mathop{\mathbb{E}}_{\mathcal{U}} \mathop{\mathbb{E}}_{(x,y)\sim P} [\mathbb{1}[\exists z \in \mathcal{U}(x) : \mathbb{A}(S_{\mathcal{U}})(z) \neq y] | (x,y) \notin S] \\
&= \mathop{\Pr}_{(x,y)\sim P}[(x,y) \notin S] \mathop{\mathbb{E}}_{(x,y)\sim P} \mathop{\Pr}_{\mathcal{U}}[\exists z \in \mathcal{U}(x) : \mathbb{A}(S_{\mathcal{U}})(z) \neq y] | (x,y) \notin S] \\
&\geq \frac{1}{3} \cdot \frac{1}{2} = \frac{1}{6}.
\end{aligned}
$$

By law of total expectation, this implies that there exists a deterministic choice of $\mathcal{U}$ such that $\mathbb{E}_{S\sim P^m} \mathrm{R}_{\mathcal{U}}(\mathbb{A}(S_{\mathcal{U}}); P) \geq \frac{1}{6}$. This establishes that $\mathbb{A}$ fails to robustly learn $\mathcal{H}$ with respect to $\mathcal{U}$ using $m$ samples.

On the other hand, $\mathcal{H}$ is robustly learnable with respect to $\mathcal{U}$ with 0 samples by means of our non-local learner $\mathbb{G}_{\mathcal{H},\mathcal{U}}$ (see Section 4 and Theorem 3) which utilizes *full* knowledge of $\mathcal{U}$. In particular, 0 samples are needed, since the graph $G_{\mathcal{H}}^{\mathcal{U}}$ will contain *no* edges by the definition of $\mathcal{H}$ (Equation 5) and $\mathcal{U}$ (Equation 6). $\qquad \square$

## 3 A global one-inclusion graph

To go beyond the limitations of local learners from Section 2, in this section, we introduce: the *global one-inclusion graph*, the main mathematical object which allows us to adopt a global perspective on robust learning. Our global one-inclusion graph is inspired by the classical one-inclusion graph introduced by [17], which leads to an algorithm that is near-optimal for (non-robust) PAC learning, and has also been adapted and used in multi-class learning [26, 11, 7] and for learning partial concept classes[1] [2]. Before introducing our global one-inclusion graph, to ease the readers, we begin first with describing the construction of the classical one-inclusion graph due to [17] and its use as a (non-robust) learner, and discuss its limitations for adversarially robust learning.

### 3.1 Background: classical one-inclusion graphs

For a given class $\mathcal{H}$ and a finite dataset $X = \{x_1, \ldots, x_n\} \subseteq \mathcal{X}$, the classical one-inclusion graph $G_{X,\mathcal{H}}$ consists of vertices $V = \{(h(x_1), \ldots, h(x_n)) : h \in \mathcal{H}\}$ where each vertex $v = (h(x_1), \ldots, h(x_n)) \in V$ is a *realizable* labeling of $X$, and two vertices $u, v \in V$ are connected with an edge if and only if they differ only in the labeling of a single $x_i \in X$. [17] showed that the edges in $G_{X,\mathcal{H}}$ can be oriented such that each vertex has out-degree at most $\mathrm{vc}(\mathcal{H})$. Now, how can the one-inclusion graph be used as a learner? Given a training set of examples $S = \{(x_1, y_1), \ldots, (x_{n-1}, y_{n-1})\}$ and a test example $x_n$, we construct the one-inclusion graph on $\{x_1, \ldots, x_{n-1}\} \cup \{x_n\}$ using the class $\mathcal{H}$ and orient it such that maximum out-degree is at most $\mathrm{vc}(\mathcal{H})$. Then, we use the orientation to predict the label of the test point $x_n$. Specifically, if there exists $h, h' \in \mathcal{H}$ such that $\forall 1 \leq i \leq n-1 : h(x_i) = h'(x_i)$ and $h(x_n) \neq h'(x_n)$ then we will have two vertices in the graph $v = (h(x_1), \ldots, h(x_{n-1}), h(x_n))$ and $u = (h(x_1), \ldots, h(x_{n-1}), h'(x_n))$ with an edge connecting them (because they differ only in the label of $x_n$), and we predict the label of $x_n$ that this edge is directed towards. Since each vertex has out-degree at most $\mathrm{vc}(\mathcal{H})$, this implies that the average leave-one-out-error (which bounds the expected risk from above) is at most $\frac{\mathrm{vc}(\mathcal{H})}{n}$.

---

[1]At a first glance, it might seem that adversarially robust learning can be viewed as a special case of learning partial concepts classes [2], but we would like to remark that this is *not* the case. The apparent similarity arises because it is possible to state the robust realizability assumption in the language of partial concept classes, as in the example mentioned in [2] on learning linear separators with a margin, but this is just an assumption on the data-distribution. Specifically, a partial concept class learner is *only* guaranteed to make few errors on samples drawn from the distribution [see Definition 2 in 2], and not on their adversarial perturbations: i.e., performance is still measured under 0-1 loss, not robust risk.

What breaks in the adversarial learning setting? At test-time, we do not observe an i.i.d. test example $x \sim \mathcal{D}$ but rather only an adversarially chosen perturbation $z \in \mathcal{U}(x)$. This completely breaks the exchangeability analysis of the classical one-inclusion graph, because the training points are i.i.d. but the perturbation $z$ of the test point $x$ is not. Furthermore, the classical one-inclusion graph is *local* in the sense that it depends on the training data and the test point, and as such different perturbations $z, \tilde{z} \in \mathcal{U}(x)$ could very well lead to different graphs, different orientations, and ultimately different predictions for $z$ and $\tilde{z}$ which by definition imply that the prediction is not robust on $x$.

### 3.2 Our global one-inclusion graph

We now describe the construction of the global one-inclusion graph. For any class $\mathcal{H}$, any perturbation set $\mathcal{U}$, and any dataset size $n \in \mathbb{N}$, denote by $G_{\mathcal{H}}^{\mathcal{U}} = (V_n, E_n)$ the global one-inclusion graph. In words, $V_n$ is the collection of all datasets of size $n$ that can be *robustly* labeled by class $\mathcal{H}$ with respect to perturbation set $\mathcal{U}$. Formally, each vertex $v \in V_n$ is represented as a *multiset* of labeled examples $(x, y)$ of size $n$:[2]

$$V_n = \left\{ \{(x_1, y_1), \ldots, (x_n, y_n)\} : (\exists h \in \mathcal{H}) \, (\forall i \in [n]) \, (\forall z \in \mathcal{U}(x_i)), \, h(z) = y_i \right\}. \qquad (7)$$

Two vertices (datasets) $u, v \in V_n$ are connected by an edge if and only if there is a unique labeled example $(x, y) \in v$ that does not appear in $u$ and there is a unique labeled example $(\tilde{x}, \tilde{y}) \in u$ that does not appear in $v$ satisfying: $y \neq \tilde{y}$ and $\mathcal{U}(x) \cap \mathcal{U}(\tilde{x}) \neq \emptyset$. Formally, $u, v \in V_n$ are connected by an edge if and only if their symmetric difference $u \Delta v = \{(x, y), (\tilde{x}, \tilde{y})\}$ where $y \neq \tilde{y}$ and $\mathcal{U}(x) \cap \mathcal{U}(\tilde{x}) \neq \emptyset$. Furthermore, we will additionally label edges by the perturbation $z \in \mathcal{U}(x) \cap \mathcal{U}(\tilde{x})$ that witnesses this edge:

$$E_n = \left\{ (\{u, v\}, z) : u, v \in V_n \wedge u \Delta v = \{(x, y), (\tilde{x}, \tilde{y})\} \wedge y \neq \tilde{y} \wedge z \in \mathcal{U}(x) \cap \mathcal{U}(\tilde{x}) \right\}. \qquad (8)$$

For each vertex $v \in V_n$, denote by $\mathrm{advdeg}(v)$ the adversarial degree of $v$ which is defined as the number of elements $(x, y) \in v$ that witness an edge incident on $v$:

$$\mathrm{advdeg}(v) = |\{(x, y) \in v : \exists u \in V_n, z \in \mathcal{X} \text{ s.t. } (\{v, u\}, z) \in E_n \wedge (x, y) \in v \Delta u\}|. \qquad (9)$$

We want to emphasize that our notion of *adversarial degree* is different from the standard notion of degree used in graph theory, and in particular different from the degree notion in the classical one-inclusion graph used above. Specifically, we do *not* count all edges incident on a vertex rather we count the number of datapoints $(x, y)$ in a vertex that witness an edge. This different notion of degree is more suitable for our purposes and is related to the average leave-one-out *robust* error.

### 3.3 From orientations to learners

An orientation $\mathcal{O} : E_n \to V_n$ of the global one-inclusion graph $G_{\mathcal{H}}^{\mathcal{U}} = (V_n, E_n)$ is a mapping that directs each edge $e = (\{u, v\}, z) \in E_n$ towards a vertex $\mathcal{O}(e) \in \{u, v\}$. Given an orientation $\mathcal{O} : E_n \to V_n$ of the global one-inclusion graph $G_{\mathcal{H}}^{\mathcal{U}}$, the adversarial out-degree of a vertex $v \in V$, denoted by $\mathrm{adv\text{-}outdeg}(v; \mathcal{O})$, is defined as the number of elements $(x, y) \in v$ that witness an out-going edge incident on $v$ according to orientation $\mathcal{O}$:

$$\mathrm{adv\text{-}outdeg}(v; \mathcal{O}) = \left| \left\{ (x, y) \in v \,\middle|\, \begin{array}{l} \exists u \in V_n, z \in \mathcal{X} \text{ s.t. } (\{v, u\}, z) \in E_n \wedge \\ (x, y) \in v \Delta u \wedge \mathcal{O}((\{v, u\}, z)) = u \end{array} \right\} \right|. \qquad (10)$$

Why are we interested in orientations of the global one-inclusion graph $G_{\mathcal{H}}^{\mathcal{U}}$? We show next that every orientation of $G_{\mathcal{H}}^{\mathcal{U}}$ can be used to construct a learner, and that the expected robust risk of this learner is bounded from above by the maximum adversarial out-degree of the corresponding orientation. We will use this observation later in Section 4 to construct an optimal learner.

**Lemma 2.** *For any class $\mathcal{H}$, any perturbation set $\mathcal{U}$, and any $n > 1$, let $G_{\mathcal{H}}^{\mathcal{U}} = (V_n, E_n)$ be the global one-inclusion graph. Then, for any orientation $\mathcal{O} : E_n \to V_n$ of $G_{\mathcal{H}}^{\mathcal{U}}$, there exists a learner $\mathbb{A}_{\mathcal{O}} : (\mathcal{X} \times \mathcal{Y})^{n-1} \to \mathcal{Y}^{\mathcal{X}}$, such that*

$$\mathcal{E}_{n-1}(\mathbb{A}_{\mathcal{O}}; \mathcal{H}, \mathcal{U}) \leq \frac{\max_{v \in V_n} \mathrm{adv\text{-}outdeg}(v; \mathcal{O})}{n}.$$

---

[2]Note that we allow a labeled example $(x, y)$ to appear more than once in a vertex $v$, hence the multiset representation.

The proof is deferred to Appendix A. At a high-level, we can use an orientation $\mathcal{O}$ of $G_{\mathcal{H}}^{\mathcal{U}}$ to make predictions in the following way: upon receiving training examples $S$ and a (possibly adversarial) test instance $z$, we consider all possible natural datapoints $(x, y)$ of which $z$ is a perturbation of $x$ (i.e., $z \in \mathcal{U}(x)$) such that $S \cup \{(x, y)\}$ can be labeled robustly using class $\mathcal{H}$ with respect to $\mathcal{U}$ (note that these are all vertices in $G_{\mathcal{H}}^{\mathcal{U}}$ by definition), and if two different robust labelings of $z$ are possible, the orientation $\mathcal{O}$ determines which label to predict. This is defined explicitly in Algorithm 1.

---

**Algorithm 1:** Converting an Orientation $\mathcal{O}$ of $G_{\mathcal{H}}^{\mathcal{U}}$ to a Learner $\mathbb{A}_{\mathcal{O}}$.

**Input:** Training dataset $S = \{(x_1, y_1), \ldots, (x_{n-1}, y_{n-1})\} \in (\mathcal{X} \times \mathcal{Y})^{n-1}$, test instance $z \in \mathcal{X}$, and an orientation $\mathcal{O} : E_n \to V_n$ of $G_{\mathcal{H}}^{\mathcal{U}} = (E_n, V_n)$.

1   Let $P_+ = \{v \in V_n : \exists x \in \mathcal{X} \text{ s.t. } z \in \mathcal{U}(x) \wedge v = \{(x_1, y_1), \ldots, (x_{n-1}, y_{n-1}), (x, +1)\}\}$.
2   Let $P_- = \{v \in V_n : \exists x \in \mathcal{X} \text{ s.t. } z \in \mathcal{U}(x) \wedge v = \{(x_1, y_1), \ldots, (x_{n-1}, y_{n-1}), (x, -1)\}\}$.
3   **If** $\left(\exists_{y \in \{\pm 1\}}\right) \left(\exists_{v \in P_y}\right) \left(\forall_{u \in P_{-y}}\right) : \mathcal{O}(\{v, u\}, z) = v$, **then** output label $y$.
4   **Otherwise**, output $+1$.

---

## 4   A generic minimax optimal learner

We now present an optimal robust learner based on our global one-inclusion graph from Section 3.

---

For any class $\mathcal{H}$, any perturbation set $\mathcal{U}$, and integer $n > 1$, let $G_{\mathcal{H}}^{\mathcal{U}} = (V_n, E_n)$ be the global one-inclusion graph (Equations 7 and 8). Let $\mathcal{O}^*$ be an orientation that minimizes the maximum adversarial out-degree of $G_{\mathcal{H}}^{\mathcal{U}}$:

$$\mathcal{O}^* \in \underset{\mathcal{O}:E_n \to V_n}{\operatorname{argmin}} \; \max_{v \in V_n} \operatorname{adv-outdeg}(v; \mathcal{O}). \tag{11}$$

Then, let $\mathbb{G}_{\mathcal{H},\mathcal{U}}$ be the learner implied by orientation $\mathcal{O}^*$ as described in Algorithm 1.

---

**Theorem 3.** *For any $\mathcal{H}, \mathcal{U}$, any $n \in \mathbb{N}$, learner $\mathbb{G}_{\mathcal{H},\mathcal{U}}$ described above satisfies for any learner $\mathbb{A}$:*

$$\mathcal{E}_{2n-1}(\mathbb{G}_{\mathcal{H},\mathcal{U}}; \mathcal{H}, \mathcal{U}) \leq 4 \cdot \mathcal{E}_n(\mathbb{A}; \mathcal{H}, \mathcal{U}), \; \& \text{ equivalently } \mathcal{M}_{\varepsilon}^{\mathrm{re}}(\mathbb{G}_{\mathcal{H},\mathcal{U}}; \mathcal{H}, \mathcal{U}) \leq 2 \cdot \mathcal{M}_{\varepsilon/4}^{\mathrm{re}}(\mathbb{A}; \mathcal{H}, \mathcal{U}) - 1.$$

Before proceeding to the proof of Theorem 3, we first prove a key Lemma which basically shows that we can use an arbitrary learner $\mathbb{A}$ to orient the edges in the global one-inclusion graph $G_{\mathcal{H}}^{\mathcal{U}}$, and that the maximum adversarial out-degree of the resultant orientation is upper bounded by the robust error rate of $\mathbb{A}$.

**Lemma 4** (Lowerbound on Error Rate of Learners). *Let $\mathbb{A} : (\mathcal{X} \times \mathcal{Y})^* \to \mathcal{Y}^{\mathcal{X}}$ be any learner, and $n \in \mathbb{N}$. Let $G_{\mathcal{H}}^{\mathcal{U}} = (V_{2n}, E_{2n})$ be the global one-inclusion graph as defined in Equation 7 and Equation 8. Then, there exists an orientation $\mathcal{O}_{\mathbb{A}} : E_{2n} \to V_{2n}$ of $G_{\mathcal{H}}^{\mathcal{U}}$ such that*

$$\mathcal{E}_n(\mathbb{A}; \mathcal{H}, \mathcal{U}) \geq \frac{1}{4} \frac{\max_{v \in V_{2n}} \operatorname{adv-outdeg}(v; \mathcal{O}_{\mathbb{A}})}{2n}.$$

*Proof.* We begin with describing the orientation $\mathcal{O}_{\mathbb{A}}$ by orienting edges incident on each vertex $v \in V_{2n}$. Consider an arbitrary vertex $v = \{(x_1, y_1), \ldots, (x_{2n}, y_{2n})\}$ and let $P_v$ be a uniform distribution over $(x_1, y_1), \ldots, (x_{2n}, y_{2n})$. For each $1 \leq t \leq 2n$, let

$$p_t(v) = \Pr_{S \sim P_v^n} \left[\exists z \in \mathcal{U}(x_t) : \mathbb{A}(S)(z) \neq y_t | (x_t, y_t) \notin S\right].$$

For each $(x_t, y_t) \in v$ that witnesses an edge, i.e. $\exists u \in V_{2n}, z \in \mathcal{X}$ s.t. $(\{v, u\}, z) \in E_{2n}$ and $(x_t, y_t) \in v \Delta u$, if $p_t < \frac{1}{2}$, then orient *all* edges incident on $(x_t, y_t)$ inward, otherwise orient them arbitrarily. Note that this might yield edges that are oriented outwards from both their endpoint vertices, in which case, we arbitrarily orient such an edge. Observe also that we will not encounter a situation where edges are oriented inwards towards both their endpoints (which is an invalid orientation). This is because for any two vertices $v, u \in V_{2n}$ such that $\exists z_0 \in \mathcal{X}$ where $(\{u, v\}, z_0) \in E_{2n}$ and $v \Delta u = \{(x_t, y_t), (\tilde{x}_t, -y_t)\}$, we cannot have $p_t(v) < \frac{1}{2}$ and $p_t(u) < \frac{1}{2}$, since

$$p_t(v) \geq \Pr_{S \sim P_v^m} \left[\mathbb{A}(S)(z_0) \neq y_t | (x_t, y_t) \notin S\right] \text{ and } p_t(u) \geq \Pr_{S \sim P_u^m} \left[\mathbb{A}(S)(z_0) \neq -y_t | (\tilde{x}_t, -y_t) \notin S\right],$$

and $P_v$ conditioned on $(x_t, y_t) \notin S$ is the same distribution as $P_u$ conditioned on $(\tilde{x}_t, -y_t) \notin S$. This concludes describing the orientation $\mathcal{O}_{\mathbb{A}}$. We now bound the adversarial out-degree of vertices $v \in V_{2n}$:

$$\text{adv-outdeg}(v; \mathcal{O}_{\mathbb{A}}) \leq \sum_{t=1}^{2n} \mathbb{1}\left[p_t \geq \frac{1}{2}\right] \leq 2 \sum_{t=1}^{2n} p_t$$

$$= 2 \sum_{t=1}^{2n} \Pr_{S \sim P^n} \left[\exists z \in \mathcal{U}(x_t) : \mathbb{A}(S)(z) \neq y_t | (x_t, y_t) \notin S\right]$$

$$= 2 \sum_{t=1}^{2n} \frac{\Pr_{S \sim P^n}\left[(\exists z \in \mathcal{U}(x_t) : \mathbb{A}(S)(z) \neq y_t) \wedge (x_t, y_t) \notin S\right]}{\Pr_{S \sim P^n}\left[(x_t, y_t) \notin S\right]}$$

$$\leq 4 \sum_{t=1}^{2n} \Pr_{S \sim P^n} \left[(\exists z \in \mathcal{U}(x_t) : \mathbb{A}(S)(z) \neq y_t) \wedge (x_t, y_t) \notin S\right]$$

$$= 4 \mathop{\mathbb{E}}_{S \sim P^n} \sum_{(x_t, y_t) \notin S} \mathbb{1}\left[\exists z \in \mathcal{U}(x_t) : \mathbb{A}(S)(z) \neq y_t\right]$$

$$\leq 4 \mathop{\mathbb{E}}_{S \sim P^n} \sum_{t=1}^{2n} \mathbb{1}\left[\exists z \in \mathcal{U}(x_t) : \mathbb{A}(S)(z) \neq y_t\right] = 8n \mathop{\mathbb{E}}_{S \sim P^n} \text{R}_{\mathcal{U}}(\mathbb{A}(S); P) \leq 8n\mathcal{E}_n(\mathbb{A}; \mathcal{H}, \mathcal{U}).$$

Since the above holds for any vertex $v \in V_{2n}$, by rearranging terms, we get $\mathcal{E}_n(\mathbb{A}; \mathcal{H}, \mathcal{U}) \geq \frac{1}{4} \frac{\max_{v \in V_{2n}} \text{adv-outdeg}(v; \mathcal{O}_{\mathbb{A}})}{2n}$. $\qquad\square$

We are now ready to proceed with the proof of Theorem 3.

*Proof of Theorem 3.* By invoking Lemma 4, we have that for any learner $\mathbb{A}$,

$$\mathcal{E}_n(\mathbb{A}; \mathcal{H}, \mathcal{U}) \geq \frac{1}{4} \frac{\max_{v \in V_{2n}} \text{adv-outdeg}(v; \mathcal{O}_{\mathbb{A}})}{2n}.$$

By Equation 11, an orientation $\mathcal{O}^*$ has smaller maximum adversarial out-degree, thus

$$\frac{1}{4} \frac{\max_{v \in V_{2n}} \text{adv-outdeg}(v; \mathcal{O}_{\mathbb{A}})}{2n} \geq \frac{1}{4} \frac{\max_{v \in V_{2n}} \text{adv-outdeg}(v; \mathcal{O}^*)}{2n}.$$

By invoking Lemma 2, it follows that our optimal learner $\mathbb{G}_{\mathcal{H}, \mathcal{U}}$ satisfies

$$\frac{1}{4} \frac{\max_{v \in V_{2n}} \text{adv-outdeg}(v; \mathcal{O}^*)}{2n} \geq \frac{\mathcal{E}_{2n-1}(\mathbb{G}_{\mathcal{H}, \mathcal{U}}; \mathcal{H}, \mathcal{U})}{4}.$$

We arrive at the theorem statement by chaining the above inequalities and rearranging terms. $\qquad\square$

## 5 A complexity measure and sample complexity bounds

In Section 4, we showed how our global one-inclusion graph yields a near-optimal learner for adversarially robust learning. We now turn to characterizing adversarially robust learnability.

Across learning theory, many fundamental learning problems can be surprisingly characterized by means of combinatorial complexity measures. Such characterizations are often quantitatively insightful in that they provide tight bounds on the number of examples needed for learning, and also insightful for algorithm design. For example, for standard (non-robust) learning, the VC dimension characterizes what classes $\mathcal{H}$ are PAC learnable [32, 33, 6, 13]. For multi-class learning, there are characterizations based on the Natarjan and Graph dimensions, and the Daniely-Shalev-Shwartz (DS) [24, 10, 7]. For learning real-valued functions, the fat-shattering dimension plays a similar role [1, 18, 29]. The Littlestone dimension characterizes online learnability [19], and the star number characterizes the label complexity of active learning [16].

[28] showed that in Vapnik's "General Learning" problem [31], the loss class having finite VC dimension is sufficient but not, in general, necessary for learnability and asked whether there is

another dimension that characterizes learnability in this setting. But recently, [4] surprisingly exhibited a statistical learning problem that can not be characterized with a combinatorial VC-like dimension. In order to do so, they presented a *formal* definition of the notion of "dimension" or "complexity measure" (see Definition 4), that all previously proposed dimensions in statistical learning theory comply with. This raises the following natural question:

*Is there a dimension that characterizes robust learnability, and if so, what is it?!*

## 5.1 A dimension characterizing robust learning

We present next a dimension for adversarially robust learnability, which is inspired by our global one-inclusion graph described in Section 3.

$$\mathfrak{D}_{\mathcal{U}}(\mathcal{H}) = \max \left\{ n \in \mathbb{N} \cup \{\infty\} \,\middle|\, \begin{array}{l} \exists \text{ a finite subgraph } G = (V, E) \text{ of } G_{\mathcal{H}}^{\mathcal{U}} = (V_n, E_n) \text{ s.t.} \\ \forall \text{ orientations } \mathcal{O} \text{ of } G, \exists v \in V \text{ where adv-outdeg}(v; \mathcal{O}) \geq \frac{n}{3}. \end{array} \right\}.$$
(12)

In Appendix D, we discuss how our dimension satisfies the formal definition proposed by [4]. We now show that $\mathfrak{D}_{\mathcal{U}}(\mathcal{H})$ characterizes robust learnability *qualitatively* and *quantitatively*.

**Theorem 5** (Qualitative Characterization). *For any class $\mathcal{H}$ and any perturbation set $\mathcal{U}$, $\mathcal{H}$ is robustly PAC learnable with respect to $\mathcal{U}$ if and only if $\mathfrak{D}_{\mathcal{U}}(\mathcal{H})$ is finite.*

**Theorem 6** (Quantitative Characterization). *For any class $\mathcal{H}$ and any perturbation set $\mathcal{U}$,*

$$\Omega \left( \frac{\mathfrak{D}_{\mathcal{U}}(\mathcal{H})}{\varepsilon} \right) \leq \mathcal{M}_{\varepsilon,\delta}^{\text{re}}(\mathcal{H}, \mathcal{U}) \leq O \left( \frac{\mathfrak{D}_{\mathcal{U}}(\mathcal{H})}{\varepsilon} \log^2 \frac{\mathfrak{D}_{\mathcal{U}}(\mathcal{H})}{\varepsilon} + \frac{\log(1/\delta)}{\varepsilon} \right).$$

Theorem 5 follows immediately from Theorem 6. To prove Theorem 6, we first prove the following key Lemma which provides upper and lower bounds on the *minimax* expected robust risk of learning a class $\mathcal{H}$ with respect to a perturbation set $\mathcal{U}$ (see Equation 3) as a function of our introduced dimension $\mathfrak{D}_{\mathcal{U}}(\mathcal{H})$. Theorem 6 follows from an argument to boost the robust risk and the confidence as appeared in [21]. The proofs are deferred to Appendix B.

**Lemma 7.** *For any class $\mathcal{H}$, any perturbation set $\mathcal{U}$, and any $\varepsilon \in (0, 1)$,*

1. $\forall n > \mathfrak{D}_{\mathcal{U}}(\mathcal{H}) : \mathcal{E}_{n-1}(\mathcal{H}, \mathcal{U}) \leq \frac{1}{3}$.

2. $\forall 2 \leq n \leq \frac{\mathfrak{D}_{\mathcal{U}}(\mathcal{H})}{2} : \mathcal{E}_{\frac{n}{\varepsilon}}(\mathcal{H}, \mathcal{U}) \geq \frac{\varepsilon}{6}$.

## 5.2 Examples

We discuss a few ways of estimating or calculating our proposed dimension $\mathfrak{D}_{\mathcal{U}}(\mathcal{H})$.

**Proposition 8.** *For any class $\mathcal{H}$ and perturbation set $\mathcal{U}$:*

$$\mathfrak{D}_{\mathcal{U}}(\mathcal{H}) \leq \min \left\{ \tilde{O}(\text{vc}(\mathcal{H})\text{vc}^*(\mathcal{H})), \tilde{O}(\text{vc}(\mathcal{L}_{\mathcal{H}}^{\mathcal{U}})) \right\},$$

*where $\text{vc}^*(\mathcal{H})$ denotes the* dual *VC dimension, and $\text{vc}(\mathcal{L}_{\mathcal{H}}^{\mathcal{U}}))$ denotes the VC dimension of the robust-loss class $\mathcal{L}_{\mathcal{H}}^{\mathcal{U}} = \left\{ (x, y) \mapsto \sup_{z \in \mathcal{U}(x)} \mathbb{1}[h(z \neq y)] : h \in \mathcal{H} \right\}$.*

*Proof.* Set $\varepsilon_0 = \frac{1}{3}$. We know from Theorem 6 that $\mathcal{M}_{\varepsilon_0}^{\text{re}}(\mathcal{H}, \mathcal{U}) \geq \Omega(\mathfrak{D}_{\mathcal{U}}(\mathcal{H}))$. We also know from [Theorem 4 in 21] that $\mathcal{M}_{\varepsilon_0}^{\text{re}}(\mathcal{H}, \mathcal{U}) \leq \tilde{O}(\text{vc}(\mathcal{H})\text{vc}^*(\mathcal{H}))$. Finally, we know from [Theorem 1 in 9] that $\mathcal{M}_{\varepsilon_0}^{\text{re}}(\mathcal{H}, \mathcal{U}) \leq \tilde{O}(\text{vc}(\mathcal{L}_{\mathcal{H}}^{\mathcal{U}}))$. Combining these together yields that stated bound. $\square$

The dual VC dimension satisfies: $\text{vc}^*(\mathcal{H}) < 2^{\text{vc}(\mathcal{H})+1}$ [3], and this exponential dependence is tight for some classes. For many natural classes, however, such as linear predictors and some neural networks [see Lemma 3.2 in 22], the primal and dual VC dimensions are equal, or at least polynomially related. Using Proposition 8, we can conclude that for such classes $\mathfrak{D}_{\mathcal{U}}(\mathcal{H}) \leq \text{poly}(\text{vc}(\mathcal{H}))$, specifically, for $\mathcal{H}$ being linear predictors, $\mathfrak{D}_{\mathcal{U}}(\mathcal{H}) \leq \tilde{O}(\text{vc}^2(\mathcal{H}))$. Furthermore, for $\mathcal{H}$ being linear predictors

and $\mathcal{U} = \ell_p$ perturbations, we know that $\text{vc}(\mathcal{L}_{\mathcal{H}}^{\mathcal{U}}) = O(\text{vc}(\mathcal{H}))$ [Theorem 2 in 9], and so using Proposition 8 again, we get a tighter bound for these $\ell_p$ perturbations $\mathfrak{D}_{\mathcal{U}}(\mathcal{H}) \leq \tilde{O}(\text{vc}(\mathcal{H}))$.

While Proposition 8 is certainly useful for estimating our dimension $\mathfrak{D}_{\mathcal{U}}(\mathcal{H})$, we get vacuous bounds when the VC dimension $\text{vc}(\mathcal{H})$ is infinite. To this end, recall the $(\mathcal{H}, \mathcal{U})$ examples in Example 1 and Example 2 mentioned in Section 1, which satisfy $\text{vc}(\mathcal{H}) = \infty$. We can calculate $\mathfrak{D}_{\mathcal{U}}(\mathcal{H})$ for these examples differently. In particular, in Example 1, by definition, the global one-inclusion graph $G_{\mathcal{H}}^{\mathcal{U}} = (V_n, E_n)$ has no edges when $n > 1$ because $\mathcal{U}(x) = \mathcal{X}$ and thus $\mathfrak{D}_{\mathcal{U}}(\mathcal{H}) \leq 1$. In Example 2, we get that $\mathfrak{D}_{\mathcal{U}}(\mathcal{H}) \leq \tilde{O}(d)$ since we can robustly learn with $O(d)$ samples, but we can also calculate $\mathfrak{D}_{\mathcal{U}}(\mathcal{H})$ directly by constructing the global one-inclusion graph $G_{\mathcal{H}}^{\mathcal{U}} = (V_n, E_n)$ for $n > 3d$ and observing that we can orient $G_{\mathcal{H}}^{\mathcal{U}}$ such that the adversarial out-degree is at most $d$, which is possible because of the definition of $\mathcal{U}$.

### 5.3 Conjectures

While we have shown that our proposed dimension $\mathfrak{D}_{\mathcal{U}}(\mathcal{H})$ in Equation 12 characterizes robust learnability, we believe that there are other *equivalent* dimensions that are *simpler* to describe. For a more appealing dimension, we may take inspiration from recent progress on multi-class learning [7], where it was shown that the DS dimension due to [11] characterizes multi-class learnability. For a class $\mathcal{H} \subseteq \mathcal{Y}^{\mathcal{X}}$ ($|\mathcal{Y}| > 2$), the DS dimension corresponds to the largest $n$ s.t. there exists points $P \in \mathcal{X}^n$ where the projection of $\mathcal{H}$ onto $P$ induces a one-inclusion hyper graph where every vertex has full-degree. This inspires the *full-degree* dimension of the global one-inclusion graph:

$$\mathfrak{FD}_{\mathcal{U}}(\mathcal{H}) = \max \left\{ n \in \mathbb{N} \cup \{\infty\} \;\middle|\; \begin{array}{l} \exists \text{ a finite subgraph } G = (V, E) \text{ of } G_{\mathcal{H}}^{\mathcal{U}} = (V_n, E_n) \text{ s.t. every} \\ \text{vertex has full-degree: } \forall v \in V, \text{advdeg}(v; E) \geq n. \end{array} \right\}.$$

This complexity measure avoids orientations, and thus, it is perhaps simpler to verify "$\mathfrak{FD}_{\mathcal{U}}(\mathcal{H}) \geq d$" than "$\mathfrak{D}_{\mathcal{U}}(\mathcal{H}) \geq d$". Furthermore, when $\mathcal{U}(x) = \{x\} \forall x \in \mathcal{X}$, the full-degree dimension, $\mathfrak{FD}_{\mathcal{U}}(\mathcal{H})$, corresponds exactly to the VC dimension of $\mathcal{H}$, $\text{vc}(\mathcal{H})$.

**Conjecture 1.** *For any class $\mathcal{H}$ and perturbation set $\mathcal{U}$, $\mathcal{M}_{\varepsilon,\delta}^{\text{re}}(\mathcal{H}, \mathcal{U}) = \Theta_{\varepsilon,\delta}(\mathfrak{FD}_{\mathcal{U}}(\mathcal{H}))$.*

[21] proposed the following combinatorial robust shattering dimension, denoted $\dim_{\mathcal{U}}(\mathcal{H})$, and showed that $\mathcal{M}_{\varepsilon,\delta}^{\text{re}}(\mathcal{H}, \mathcal{U}) \geq \Omega_{\varepsilon,\delta}(\dim_{\mathcal{U}}(\mathcal{H}))$.

**Definition 2** (Robust Shattering Dimension). *A sequence $z_1, \ldots, z_k \in \mathcal{X}$ is said to be $\mathcal{U}$-robustly shattered by $\mathcal{H}$ if $\exists x_1^+, x_1^-, \ldots, x_k^+, x_k^- \in \mathcal{X}$ s.t. $\forall i \in [k], z_i \in \mathcal{U}(x_i^+) \cap \mathcal{U}(x_i^-)$ and $\forall y_1, \ldots, y_k \in \{\pm 1\} : \exists h \in \mathcal{H}$ such that $h(z') = y_i \forall z' \in \mathcal{U}(x_i^{y_i}) \forall 1 \leq i \leq k$. The $\mathcal{U}$-robust shattering dimension $\dim_{\mathcal{U}}(\mathcal{H})$ is defined as the largest $k$ for which there exist $k$ points $\mathcal{U}$-robustly shattered by $\mathcal{H}$.*

In regards to the relationship between the robust shattering dimension $\dim_{\mathcal{U}}(\mathcal{H})$ and our dimension $\mathfrak{D}_{\mathcal{U}}(\mathcal{H})$, we conjecture that our dimension can be arbitrarily larger. In other words, we conjecture that the robust shattering dimension $\dim_{\mathcal{U}}(\mathcal{H})$ does not characterize robust learnability.

**Conjecture 2.** *$\forall n \in \mathbb{N}, \exists \mathcal{X}, \mathcal{H}, \mathcal{U}$, such that $\dim_{\mathcal{U}}(\mathcal{H}) = O(1)$ but $\mathfrak{D}_{\mathcal{U}}(\mathcal{H}) \geq n$.*

We find this to be analogous to a separation in multi-class learnability, where the Natarajan dimension was shown to not characterize multi-class learnability [7]. Because in both graphs, the one-inclusion hyper graph and our global one-inclusion graph, the Natarajan and robust shattering dimensions represent a "cube" in their corresponding graph, while the DS dimension and our full-degree dimension represent a "pseudo-cube" in the terminology of [7]. To elaborate, the robust shattering dimension (Definition 2) gives rise to a "cube" in our global one-inclusion graph, meaning it is a subgraph isomorphic to the Boolean cube $\{0, 1\}^n$. Specifically, any $z_1, \ldots, z_k$ that are robustly shattered (as in Definition 2) can be used to construct a finite subgraph where the vertices are $\{(x_1^{y_1}), \ldots, (x_n^{y_n})\}$ for $y \in \{\pm 1\}$. By our definition of degree, these vertices will have full-degree, since every vertex will have a neighbour (with only a single label flipped). In contrast, the full-degree dimension that we propose (see Conjecture 1 above) gives any finite subgraph where every vertex has full-degree, including those not isomorphic to the Boolean cube. Following the terminology in multiclass learning, we call this a "pseudo-cube", as it needn't be isomorphic to the Boolean cube.

Another interesting and perhaps useful direction to explore is the relationship between our proposed complexity measure $\mathfrak{D}_{\mathcal{U}}(\mathcal{H})$ and the VC dimension. We believe that it is actually possible to orient the global one-inclusion graph such that the maximum adversarial out-degree is at most $O(\text{vc}(\mathcal{H}))$.

**Conjecture 3.** *For any class $\mathcal{H}$ and perturbation set $\mathcal{U}$, $\mathfrak{D}_{\mathcal{U}}(\mathcal{H}) \leq O(\text{vc}(\mathcal{H}))$.*

## 6 Agnostic robust learnability

For the agnostic setting, we consider robust learnability with respect to arbitrary distributions $\mathcal{D}$ that are *not* necessarily robustly realizable, i.e., $\mathcal{D} \notin \mathrm{RE}(\mathcal{H}, \mathcal{U})$ (see Definition 3 in Appendix C). We can establish an upper bound in the agnostic setting via reduction to the realizable case, following an argument from [12] and later applied to agnostic robust learning by [21]:

**Theorem 9.** *For any class $\mathcal{H}$ and any perturbation set $\mathcal{U}$,*

$$\mathcal{M}^{\mathrm{ag}}_{\varepsilon,\delta}(\mathcal{H}, \mathcal{U}) = O\left(\frac{\mathfrak{D}_{\mathcal{U}}(\mathcal{H})}{\varepsilon^2}\log^2\left(\frac{\mathfrak{D}_{\mathcal{U}}(\mathcal{H})}{\varepsilon}\right) + \frac{1}{\varepsilon^2}\log\left(\frac{1}{\delta}\right)\right).$$

This is achieved by applying the agnostic-to-realizable reduction to the optimal learner $\mathbb{G}_{\mathcal{H},\mathcal{U}}$ that we get from orienting the graph $G^{\mathcal{U}}_{\mathcal{H}} = (V_{\mathfrak{D}_{\mathcal{U}}(\mathcal{H})+1}, E_{\mathfrak{D}_{\mathcal{U}}(\mathcal{H})+1})$. The reduction is stated abstractly in the following Lemma whose proof is provided in Appendix C.

**Lemma 10.** *For any well-defined realizable learner $\mathbb{A}$, there is an agnostic learner $\mathbb{B}$ such that*

$$\mathcal{M}^{\mathrm{re}}_{\varepsilon}(\mathbb{A}; \mathcal{H}, \mathcal{U}) \leq \mathcal{M}^{\mathrm{ag}}_{\varepsilon,\delta}(\mathbb{B}; \mathcal{H}, \mathcal{U}) \leq O\left(\frac{\mathcal{M}^{\mathrm{re}}_{1/3}(\mathbb{A}; \mathcal{H}, \mathcal{U})}{\varepsilon^2}\log^2\left(\frac{\mathcal{M}^{\mathrm{re}}_{1/3}(\mathbb{A}; \mathcal{H}, \mathcal{U})}{\varepsilon}\right) + \frac{1}{\varepsilon^2}\log\left(\frac{1}{\delta}\right)\right).$$

Theorem 9 immediately follows by combining Lemma 10 and Theorem 6.

## Acknowledgments and Disclosure of Funding

We thank Shay Moran for numerous enlightening discussions about structures arising in the analysis of muliclass learning, concurrent to this work, which seem tantalizingly related to those arising here. This work is supported in part by DARPA under cooperative agreement HR00112020003 [3], and in part by the NSF/Simons sponsored Collaboration on the Theoretical Foundations of Deep Learning (https://deepfoundations.ai).

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
