# A  Proof of Lemma 2

*Proof.* Let $\mathcal{O} : E_n \to V_n$ be an arbitrary orientation of $G_{\mathcal{H}}^{\mathcal{U}}$. We will show that orientation $\mathcal{O}$ implies a learner $\mathbb{A}_{\mathcal{O}} : (\mathcal{X} \times \mathcal{Y})^{n-1} \to \mathcal{Y}^{\mathcal{X}}$ with an expected robust risk $\mathcal{E}_{n-1}$ that is upper bounded by the maximum adversarial out-degree of orientation $\mathcal{O}$.

We begin with describing the learner $\mathbb{A}_{\mathcal{O}}$. For each input $((x_1, y_1), \dots, (x_{n-1}, y_{n-1}), z) \in (\mathcal{X} \times \mathcal{Y})^{n-1} \times \mathcal{X}$ define $\mathbb{A}_{\mathcal{O}}((x_1, y_1), \dots, (x_{n-1}, y_{n-1}))(z)$ as follows. Consider the set of vertices $v \in V$ that have the multiset $\{(x_1, y_1), \dots, (x_{n-1}, y_{n-1})\}$ and perturbation $z$ with a positive label

$$P_+ = \{v \in V : \exists x \in \mathcal{X} \text{ s.t. } z \in \mathcal{U}(x) \wedge v = \{(x_1, y_1), \dots, (x_{n-1}, y_{n-1}), (x, +1)\}\},$$

and the set of vertices $v \in V$ that have the multiset $\{(x_1, y_1), \dots, (x_{n-1}, y_{n-1})\}$ and perturbation $z$ with a negative label

$$P_- = \{v \in V : \exists x \in \mathcal{X} \text{ s.t. } z \in \mathcal{U}(x) \wedge v = \{(x_1, y_1), \dots, (x_{n-1}, y_{n-1}), (x, -1)\}\}.$$

We define $\mathbb{A}_{\mathcal{O}}((x_1, y_1), \dots, (x_{n-1}, y_{n-1}))(z)$ as a function of $P_+$, $P_-$, and the orientation $\mathcal{O}$:

$$\mathbb{A}_{\mathcal{O}}((x_1, y_1), \dots, (x_{n-1}, y_{n-1}))(z) = \begin{cases} y & \text{if } \left(\exists_{y \in \{\pm 1\}}\right) \left(\exists_{v \in P_y}\right) \left(\forall_{u \in P_{-y}}\right) : \mathcal{O}((\{v, u\}, z)) = v. \\ +1 & \text{if } P_+ \neq \emptyset \wedge P_- = \emptyset. \\ -1 & \text{if } P_+ = \emptyset \wedge P_- \neq \emptyset. \\ +1 & \text{otherwise.} \end{cases}$$

Note that $\mathbb{A}_{\mathcal{O}}$ is well-defined. Specifically, observe that when $P_+ \neq \emptyset$ and $P_- \neq \emptyset$, by definition of $P_+$ and $P_-$ and Equation 8, vertices from $P_+$ and $P_-$ form a complete bipartite graph. That is, for each $v \in P_+$ and each $u \in P_-$, $(\{u, v\}, z) \in E$. This implies that there exists at most one label: either $y = +1$ or $y = -1$ such that there is a vertex $v \in P_y$ where all edges $(\{v, u\}, z) \in E$ for $u \in P_{-y}$ are incident on $v$ according to orientation $\mathcal{O}$: $(\exists! y \in \{\pm 1\}) (\exists v \in P_y) (\forall u \in P_{-y}) : \mathcal{O}((\{v, u\}, z)) = v$.

We now proceed with bounding from above the expected robust risk $\mathcal{E}_{n-1}$ of learner $\mathbb{A}_{\mathcal{O}}$ by the maximum adversarial out-degree of orientation $\mathcal{O}$. Consider an arbitrary multiset $S = \{(x_1, y_1), \dots, (x_n, y_n)\} \in (\mathcal{X} \times \mathcal{Y})^n$ that is *robustly* realizable with respect to $(\mathcal{H}, \mathcal{U})$. By definition $S \in V_n$, i.e., $S$ is a vertex in $G_{\mathcal{H}}^{\mathcal{U}}$. By definition of adversarial out-degree (see Equation 10), there exists $T \subseteq S$ where $|T| = n - \text{adv-outdeg}(S; \mathcal{O})$ such that for each $(x, y) \in T$ and for each $z \in \mathcal{U}(x)$: vertex $S$ will satisfy the condition that if there is any other vertex $u \in V_n$ where $(\{S, u\}, z)$ is an edge: $(\{S, u\}, z) \in E_n$, the orientation of edge $(\{S, u\}, z)$ is towards $S$: $\mathcal{O}((\{S, u\}, z)) = S$. Thus by definition of $\mathbb{A}_{\mathcal{O}}$ above, $\mathbb{A}_{\mathcal{O}}(S \setminus \{(x, y)\}, z) = y$. This implies that

$$\frac{1}{n} \sum_{i=1}^n \mathbb{1}\left[\exists z \in \mathcal{U}(x_i) : \mathbb{A}_{\mathcal{O}}(S \setminus \{(x_i, y_i)\})(z) \neq y_i\right] = \frac{\text{adv-outdeg}(S; \mathcal{O})}{n}.$$

To conclude, by definition of $\mathcal{E}_{n-1}(\mathbb{A}_{\mathcal{O}}; \mathcal{H}, \mathcal{U})$ (see Equation 2),

$$\begin{aligned} \mathcal{E}_{n-1}(\mathbb{A}_{\mathcal{O}}; \mathcal{H}, \mathcal{U}) &= \sup_{\mathcal{D} \in \text{RE}(\mathcal{H}, \mathcal{U})} \mathbb{E}_{S \sim \mathcal{D}^{n-1}} R_{\mathcal{U}}(\mathbb{A}_{\mathcal{O}}(S); \mathcal{D}) \\ &= \sup_{\mathcal{D} \in \text{RE}(\mathcal{H}, \mathcal{U})} \mathbb{E}_{S \sim \mathcal{D}^{n-1}} \mathbb{E}_{(x, y) \sim \mathcal{D}} \mathbb{1}\{\exists z \in \mathcal{U}(x) : \mathbb{A}_{\mathcal{O}}(S)(z) \neq y\} \\ &= \sup_{\mathcal{D} \in \text{RE}(\mathcal{H}, \mathcal{U})} \mathbb{E}_{S \sim \mathcal{D}^n} \frac{1}{n} \sum_{i=1}^n \mathbb{1}\{\exists z \in \mathcal{U}(x_i) : \mathbb{A}_{\mathcal{O}}(S \setminus \{(x_i, y_i)\})(z) \neq y_i\} \\ &\leq \frac{\max_{v \in V_n} \text{adv-outdeg}(v; \mathcal{O})}{n}. \end{aligned}$$

$\square$

# B  Lemmas and Proofs for Theorem 6

**Lemma 11** (Rado's Selection Principle [25, 34])**.** *Let $I$ be an arbitrary index set, and let $\{X_i : i \in I\}$ be a family of non-empty finite sets. For each finite subset $A$ of $I$, let $f_A$ be a choice function whose domain is $A$ and such that $f_A(i) \in X_i$ for each $i \in A$. Then, there exists a choice function $f$ whose domain is $I$ with the following property: to every finite subset $A$ of $I$ there corresponds a finite set $B$, $A \subseteq B \subseteq I$, with $f(i) = f_B(i)$ for each $i \in A$.*

**Lemma 12** (Refined Lowerbound on Error Rate of Learners). *For any integer $n \geq 1$, let $G_{\mathcal{H}}^{\mathcal{U}} = (V_{2n}, E_{2n})$ be the global one-inclusion graph as defined in Equation 7 and Equation 8. Then, for any learner $\mathbb{A} : (\mathcal{X} \times \mathcal{Y})^* \to \mathcal{Y}^{\mathcal{X}}$ and any $\varepsilon \in (0, 1)$, there exists an orientation $\mathcal{O}_{\mathbb{A}} : E_{2n} \to V_{2n}$ of $G_{\mathcal{H}}^{\mathcal{U}}$ such that*

$$\mathcal{E}_{\frac{n}{\epsilon}}(\mathbb{A}; \mathcal{H}, \mathcal{U}) \geq \frac{\varepsilon}{2} \cdot \frac{\max_{v \in V_{2n}} \text{adv-outdeg}(v; \mathcal{O}_{\mathbb{A}}) - 1}{n - 1}.$$

*of Lemma 12.* Set $m = \frac{n}{\varepsilon}$. We begin with describing the orientation $\mathcal{O}_{\mathbb{A}}$ by orienting edges incident on each vertex $v \in V_{2n}$. Consider an arbitrary vertex $v = \{(x_1, y_1), \ldots, (x_{2n}, y_{2n})\}$. Without loss of generality, let $P_v$ be a distribution over $\{(x_1, y_1), \ldots, (x_{2n}, y_{2n})\}$, defined as

$$P_v(\{(x_1, y_1)\}) = 1 - \varepsilon \text{ and } P_v(\{(x_t, y_t)\}) = \frac{\varepsilon}{2n - 1} \quad \forall 2 \leq t \leq 2n.$$

For each $1 \leq t \leq 2n$, let

$$p_t(v) = \Pr_{S \sim P_v^m} [\exists z \in \mathcal{U}(x_t) : \mathbb{A}(S)(z) \neq y_t | (x_t, y_t) \notin S].$$

For each $1 \leq t \leq 2n$ such that $(x_t, y_t) \in v$ witnesses an edge, i.e. $\exists u \in V_{2n}, z \in \mathcal{X}$ s.t. $(\{v, u\}, z) \in E_{2n}$ and $(x_t, y_t) \in v\Delta u$, if $p_t < \frac{1}{2}$, then orient *all* edges incident on $(x_t, y_t)$ inward, otherwise orient them arbitrarily. Note that this might yield edges that are oriented outwards from both their endpoint vertices, in which case, we arbitrarily orient such an edge. Observe also that we will not encounter a situation where edges are oriented inwards towards both their endpoints (which is an invalid orientation). This is because for any two vertices $v, u \in V_{2n}$ such that $\exists z_0 \in \mathcal{X}$ where $(\{u, v\}, z_0) \in E_{2n}$ and $v\Delta u = \{(x_t, y_t), (\tilde{x}_t, -y_t)\}$, we can not have $p_t(v) < \frac{1}{2}$ and $p_t(u) < \frac{1}{2}$, since

$$p_t(v) \geq \Pr_{S \sim P_v^m} [\mathbb{A}(S)(z_0) \neq y_t | (x_t, y_t) \notin S] \text{ and } p_t(u) \geq \Pr_{S \sim P_u^m} [\mathbb{A}(S)(z_0) \neq -y_t | (\tilde{x}_t, -y_t) \notin S],$$

and $P_v$ conditioned on $(x_t, y_t) \notin S$ is the same distribution as $P_u$ conditioned on $(\tilde{x}_t, -y_t) \notin S$. This concludes describing the orientation $\mathcal{O}_{\mathbb{A}}$. We now bound from above the adversarial out-degree of vertices $v \in V_{2n}$ with respect to the orientation $\mathcal{O}_{\mathbb{A}}$:

$$\text{adv-outdeg}(v; \mathcal{O}_{\mathbb{A}}) \leq \sum_{t=1}^{2n} \mathbb{1}\left[p_t \geq \frac{1}{2}\right] \leq 1 + \sum_{t=2}^{2n} \mathbb{1}\left[p_t \geq \frac{1}{2}\right] \leq 1 + 2\sum_{t=2}^{2n} p_t$$

$$= 1 + 2\sum_{t=2}^{2n} \Pr_{S \sim P^m} [\exists z \in \mathcal{U}(x_t) : \mathbb{A}(S)(z) \neq y_t | (x_t, y_t) \notin S]$$

$$= 1 + 2\sum_{t=2}^{2n} \frac{\Pr_{S \sim P^m} [(\exists z \in \mathcal{U}(x_t) : \mathbb{A}(S)(z) \neq y_t) \wedge (x_t, y_t) \notin S]}{\Pr_{S \sim P^m} [(x_t, y_t) \notin S]}$$

$$\overset{(i)}{\leq} 1 + 2 \cdot \left(1 - \frac{n}{2n - 1}\right) \sum_{t=2}^{2n} \Pr_{S \sim P^m} [(\exists z \in \mathcal{U}(x_t) : \mathbb{A}(S)(z) \neq y_t) \wedge (x_t, y_t) \notin S]$$

$$= 1 + 2 \cdot \left(1 - \frac{n}{2n - 1}\right) \sum_{t=2}^{2n} \mathbb{E}_{S \sim P^m} [\mathbb{1}[\exists z \in \mathcal{U}(x_t) : \mathbb{A}(S)(z) \neq y_t] \mathbb{1}[(x_t, y_t) \notin S]]$$

$$= 1 + 2 \cdot \left(1 - \frac{n}{2n - 1}\right) \mathbb{E}_{S \sim P^m} \left[\sum_{t=2}^{2n} \mathbb{1}[\exists z \in \mathcal{U}(x_t) : \mathbb{A}(S)(z) \neq y_t] \mathbb{1}[(x_t, y_t) \notin S]\right]$$

$$\leq 1 + 2 \cdot \left(1 - \frac{n}{2n - 1}\right) \mathbb{E}_{S \sim P^m} \left[\sum_{t=2}^{2n} \mathbb{1}[\exists z \in \mathcal{U}(x_t) : \mathbb{A}(S)(z) \neq y_t]\right]$$

$$= 1 + 2 \cdot \left(1 - \frac{n}{2n - 1}\right) \cdot \frac{2n - 1}{\varepsilon} \mathbb{E}_{S \sim P^m} \left[\frac{\varepsilon}{2n - 1} \sum_{t=2}^{2n} \mathbb{1}[\exists z \in \mathcal{U}(x_t) : \mathbb{A}(S)(z) \neq y_t]\right]$$

$$\leq 1 + 2 \cdot \left(1 - \frac{n}{2n - 1}\right) \cdot \frac{2n - 1}{\varepsilon} \mathbb{E}_{S \sim P^m} R_{\mathcal{U}}(\mathbb{A}(S); P)$$

$$\leq 1 + 2 \cdot \left(1 - \frac{n}{2n - 1}\right) \cdot \frac{2n - 1}{\varepsilon} \mathcal{E}_m(\mathbb{A}; \mathcal{H}, \mathcal{U}) = 1 + \frac{2(n - 1)}{\varepsilon} \mathcal{E}_m(\mathbb{A}; \mathcal{H}, \mathcal{U}),$$

where inequality $(i)$ follows from the following:

$$\Pr_{S \sim P^m}[(x_t, y_t) \notin S] = \left(1 - \frac{\varepsilon}{2n-1}\right)^m \geq 1 - m \cdot \frac{\varepsilon}{2n-1} = 1 - \frac{n}{\varepsilon}\frac{\varepsilon}{2n-1} \geq 1 - \frac{n}{2n-1}.$$

Since the above holds for any vertex $v \in V_{2n}$, by rearranging terms, we get $\mathcal{E}_m(\mathbb{A}; \mathcal{H}, \mathcal{U}) \geq \frac{\varepsilon}{2}\frac{\max_{v \in V_{2n}}\text{adv-outdeg}(v; \mathcal{O}_{\mathbb{A}})-1}{n-1}$. $\qquad\square$

*Proof of Lemma 7.* We will first start with the upper bound. Let $n > \mathfrak{D}_{\mathcal{U}}(\mathcal{H})$ and let $G_{\mathcal{H}}^{\mathcal{U}} = (V_n, E_n)$ be the (possibly infinite) one-inclusion graph. Then, by definition of $\mathfrak{D}_{\mathcal{U}}(\mathcal{H})$, for every finite subgraph $G = (V, E)$ of $G_{\mathcal{H}}^{\mathcal{U}}$ there exists an orientation $\mathcal{O}_E : E \to V$ such that every vertex in the subgraph has adversarial out-degree at most $\frac{n}{3}$: $\forall v \in V, \text{adv-outdeg}(v; \mathcal{O}_E) \leq \frac{n}{3}$.

We next invoke Lemma 11 where $E_n$ represents our family of non-empty finite sets, and for each finite subset $E \subseteq E_n$, we let the orientation $\mathcal{O}_E$ (from above) represent the choice function. Then, Lemma 11 implies that there exists an orientation $\mathcal{O} : E_n \to V_n$ of $G_{\mathcal{H}}^{\mathcal{U}}$ (i.e., an orientation of the entire global one-inclusion graph) with the following property: for each finite subset $A$ of $E_n$, there corresponds a finite set $E$ satisfying $A \subseteq E \subseteq E_n$ and $\mathcal{O}(e) = \mathcal{O}_E(e)$ for each $e \in A$. This implies that orientation $\mathcal{O}$ satisfies the property that $\forall v \in V_n, \text{adv-outdeg}(v; \mathcal{O}) \leq \frac{n}{3}$. Because, if not, then we can find a subgraph $G = (E, V)$ where $\mathcal{O}_E$ (from above) violates the adversarial out-degree upper bound of $\frac{n}{3}$ and that leads to a contradiction.

Now, we use orientation $\mathcal{O}$ of $G_{\mathcal{H}}^{\mathcal{U}}$ (which has adversarial out-degree at most $\frac{n}{3}$) to construct a learner $\mathbb{A}_{\mathcal{O}} : (\mathcal{X} \times \mathcal{Y})^{n-1} \times \mathcal{X} \to \mathcal{Y}$ as in Lemma 2. Then, Lemma 2 implies that

$$\mathcal{E}_{n-1}(\mathcal{H}, \mathcal{U}) \leq \mathcal{E}_{n-1}(\mathbb{A}_{\mathcal{O}}; \mathcal{H}, \mathcal{U}) \leq \frac{1}{3}.$$

We now turn to the lower bound. Let $2 \leq n \leq \frac{\mathfrak{D}_{\mathcal{U}}(\mathcal{H})}{2}$, $\varepsilon \in (0, 1)$, and let $G_{\mathcal{H}}^{\mathcal{U}} = (V_{2n}, E_{2n})$ be the (possibly infinite) one-inclusion graph. Since $2n \leq \mathfrak{D}_{\mathcal{U}}(\mathcal{H})$, by definition of $\mathfrak{D}_{\mathcal{U}}(\mathcal{H})$, it follows that there exists a finite subgraph $G = (V, E)$ of $G_{\mathcal{H}}^{\mathcal{U}} = (V_{2n}, E_{2n})$ such that

$$\forall \text{ orientations } \mathcal{O} : E \to V \text{ of subgraph } G, \max_{v \in V}\text{adv-outdeg}(v; \mathcal{O}) \geq \frac{2n}{3}. \qquad (13)$$

Now, let $\mathbb{A} : (\mathcal{X} \times \mathcal{Y})^* \to \mathcal{Y}^{\mathcal{X}}$ be an arbitrary learner. We invoke Lemma 12, which is a refined statement of Lemma 4 that takes $\varepsilon$ into account, to orient the subgraph $G$ using learner $\mathbb{A}$. Lemma 12 and Equation 13 above imply that

$$\mathcal{E}_{\frac{n}{\varepsilon}}(\mathcal{H}, \mathcal{U}) \geq \mathcal{E}_{\frac{n}{\varepsilon}}(\mathbb{A}; \mathcal{H}, \mathcal{U}) \geq \frac{\varepsilon}{2}\frac{\max_{v \in V}\text{adv-outdeg}(v; \mathcal{O}_{\mathbb{A}}) - 2}{n-1} \geq \frac{\varepsilon}{2}\frac{(2n)/3 - 1}{n-1}$$

$$= \frac{\varepsilon}{3}\frac{2n - 2 - 1}{2n - 2} = \frac{\varepsilon}{3}\left(1 - \frac{1}{2n - 2}\right) \geq \frac{\varepsilon}{6}.$$

$\qquad\square$

**Lemma 13** (Sample Compression Robust Generalization – [21]). *For any $k \in \mathbb{N}$ and fixed function $\phi : (\mathcal{X} \times \mathcal{Y})^k \to \mathcal{Y}^{\mathcal{X}}$, for any distribution $P$ over $\mathcal{X} \times \mathcal{Y}$ and any $m \in \mathbb{N}$, for $S = \{(x_1, y_1), \dots, (x_m, y_m)\}$ iid P-distributed random variables, with probability at least $1 - \delta$, if $\exists i_1, \dots, i_k \in \{1, \dots, m\}$ s.t. $\hat{R}_{\mathcal{U}}(\phi((x_{i_1}, y_{i_1}), \dots, (x_{i_k}, y_{i_k})); S) = 0$, then*

$$R_{\mathcal{U}}(\phi((x_{i_1}, y_{i_1}), \dots, (x_{i_k}, y_{i_k})); P) \leq \frac{1}{m-k}(k \ln(m) + \ln(1/\delta)).$$

We are now ready to proceed with the proof of Theorem 6.

*Proof of Theorem 6.* We begin with proving the upper bound. Let $m_0 = \mathfrak{D}_{\mathcal{U}}(\mathcal{H})$. By Lemma 7, there exists a learner $\mathbb{A}$ from orienting the global one-inclusion graph $G_{\mathcal{H}}^{\mathcal{U}} = (V_{m_0+1}, E_{m_0+1})$ that satisfies worst-case expected risk

$$\mathcal{E}_{m_0}(\mathbb{A}; \mathcal{H}, \mathcal{U}) \leq \frac{1}{3}. \qquad (14)$$

Let $\mathcal{D} \in \text{RE}(\mathcal{H}, \mathcal{U})$ be some unknown *robustly realizable* distribution. Fix $\varepsilon, \delta \in (0, 1)$ and a sample size $m(\varepsilon, \delta)$ that will be determined later. Let $S = \{(x_1, y_1), \dots, (x_m, y_m)\}$ be an i.i.d. sample from $\mathcal{D}$. Our strategy is to use $\mathbb{A}$ above as a *weak* robust learner and boost its confidence and robust error guarantee.

**Weak Robust Learner.** Observe that, by Equation 14, for any empirical distribution $D$ over $S$, $\mathbb{E}_{S' \sim D^{m_0}} \mathrm{R}_{\mathcal{U}}(\mathbb{A}(S'); D) \leq 1/3$. This implies that for any empirical distribution $D$ over $S$, there exists at least one sequence $S_D \in (S)^{m_0}$ such that $h_D := \mathbb{A}(S_D)$ satisfies $\mathrm{R}_{\mathcal{U}}(h_D; D) \leq 1/3$. We use this to define a weak robust-learner for distributions $D$ over $S$: i.e., for any $D$, the weak learner chooses $h_D$ as its weak hypothesis.

**Boosting.** Now we run the $\alpha$-Boost boosting algorithm [27, Section 6.4.2] on data set $S$, but using the robust loss rather than 0-1 loss. That is, we start with $D_1$ uniform on $S$. Then for each round $t$, we get $h_{D_t}$ as a weak robust classifier with respect to $D_t$, and for each $(x, y) \in S$ we define a distribution $D_{t+1}$ over $S$ satisfying

$$D_{t+1}(\{(x,y)\}) = \frac{D_t(\{(x,y)\})}{Z_t} \times \begin{cases} e^{-2\alpha} & \text{if } \mathbb{1}[\forall z \in \mathcal{U}(x) : h_{D_t}(z) = y] = 1 \\ 1 & \text{otherwise} \end{cases}$$

where $Z_t$ is a normalization factor, $\alpha$ is a parameter that will be determined below. Following the argument from [27, Section 6.4.2], after $T$ rounds we are guaranteed

$$\min_{(x,y) \in S} \frac{1}{T} \sum_{t=1}^{T} \mathbb{1}[\forall z \in \mathcal{U}(x) : h_{D_t}(z) = y] \geq \frac{2}{3} - \frac{2}{3}\alpha - \frac{\ln(|S|)}{2\alpha T},$$

so we will plan on running until round $T = 1 + 48 \ln(|S|)$ with value $\alpha = 1/8$ to guarantee

$$\min_{(x,y) \in S} \frac{1}{T} \sum_{t=1}^{T} \mathbb{1}[\forall z \in \mathcal{U}(x) : h_{D_t}(z) = y] > \frac{1}{2},$$

so that the majority-vote classifier $\mathrm{MAJ}(h_{D_1}, \ldots, h_{D_T})$ achieves *zero* robust loss on the empirical dataset $S$, $\mathrm{R}_{\mathcal{U}}(\mathrm{MAJ}(h_{D_1}, \ldots, h_{D_T}); S) = 0$.

Furthermore, note that, since each $h_{D_t}$ is given by $\mathbb{A}(S_{D_t})$, where $S_{D_t}$ is an $m_0$-tuple of points in $S$, the classifier $\mathrm{MAJ}(h_{D_1}, \ldots, h_{D_T})$ is specified by an ordered sequence of $m_0 \cdot T$ points from $S$. Thus, the classifier $\mathrm{MAJ}(h_1, \ldots, h_T)$ is representable as the value of an (order-dependent) reconstruction function $\phi$ with a compression set size $m_0 T = m_0 O(\log m)$. Now, invoking Lemma 13, we get the following *robust* generalization guarantee: with probability at least $1 - \delta$ over $S \sim \mathcal{D}^m$,

$$\mathrm{R}_{\mathcal{U}}(\mathrm{MAJ}(h_1, \ldots, h_T); \mathcal{D}) \leq O\left(\frac{m_0 \log^2 m}{m} + \frac{\log(1/\delta)}{m}\right),$$

and setting this less than $\varepsilon$ and solving for a sufficient size of $m$ yields the stated sample complexity bound.

We now turn to proving the lower bound. Let $n_0 = \frac{\mathfrak{D}_{\mathcal{U}}(\mathcal{H})}{2}$, by invoking Lemma 7, we get that $\mathcal{E}_{n_0/\varepsilon} \geq \Omega(\varepsilon)$. Then, by Equation 4, this implies that $\mathcal{M}_\varepsilon(\mathcal{H}, \mathcal{U}) \geq \Omega(1/\varepsilon)n_0 \geq \Omega(1/\varepsilon)\mathfrak{D}_{\mathcal{U}}(\mathcal{H})$. □

## C  Proofs for Section 6

**Definition 3** (Agnostic Robust PAC Learnability). For any $\varepsilon, \delta \in (0, 1)$, the *sample complexity of agnostic robust $(\varepsilon, \delta)-$PAC learning of $\mathcal{H}$ with respect to perturbation set $\mathcal{U}$*, denoted $\mathcal{M}^{\mathrm{ag}}_{\varepsilon,\delta}(\mathcal{H}, \mathcal{U})$, is defined as the smallest $m \in \mathbb{N} \cup \{0\}$ for which there exists a learner $\mathbb{A} : (\mathcal{X} \times \mathcal{Y})^* \to \mathcal{Y}^{\mathcal{X}}$ such that, for every data distribution $\mathcal{D}$ over $\mathcal{X} \times \mathcal{Y}$, with probability at least $1 - \delta$ over $S \sim \mathcal{D}^m$,

$$\mathrm{R}_{\mathcal{U}}(\mathbb{A}(S); \mathcal{D}) \leq \inf_{h \in \mathcal{H}} \mathrm{R}_{\mathcal{U}}(h; \mathcal{D}) + \varepsilon.$$

If no such $m$ exists, define $\mathcal{M}^{\mathrm{ag}}_{\varepsilon,\delta}(\mathcal{H}, \mathcal{U}) = \infty$. We say that $\mathcal{H}$ is robustly PAC learnable in the agnostic setting with respect to perturbation set $\mathcal{U}$ if $\forall \epsilon, \delta \in (0, 1)$, $\mathcal{M}^{\mathrm{ag}}_{\varepsilon,\delta}(\mathcal{H}, \mathcal{U})$ is finite.

*Proof of Lemma 10.* The argument follows closely a proof of an analogous result by [12] for non-robust learning, and [21] for robust learning. Denote by $\mathbb{A}$ a realizable learner with sample complexity $\mathcal{M}^{\mathrm{re}}_{1/3}(\mathcal{H}, \mathcal{U})$, and denote $m_0 = \mathcal{M}^{\mathrm{re}}_{1/3}(\mathcal{H}, \mathcal{U})$.

**Description of agnostic learner** $\mathbb{B}$. Given a data set $S \sim \mathcal{D}^m$ where $\mathcal{D}$ is some unknown distribution, we first do robust-ERM to find a maximal-size subsequence $S'$ of the data where the robust loss can be zero: that is, $\inf_{h \in \mathcal{H}} \hat{R}_{\mathcal{U}}(h; S') = 0$. Then for any distribution $D$ over $S'$, there exists a sequence $S_D \in (S')^{m_0}$ such that $h_D := \mathbb{A}(S_D)$ has $R_{\mathcal{U}}(h_D; D) \leq 1/3$; this follows since, by definition of $\mathcal{M}_{1/3}^{\mathrm{re}}(\mathcal{H}, \mathcal{U})$, $\mathcal{E}_{m_0}(\mathbb{A}; \mathcal{H}, \mathcal{U}) \leq 1/3$ so at least one such $S_D$ exists. We use this to define a weak robust-learner for distributions $D$ over $S'$: i.e., for any $D$, the weak learner chooses $h_D$ as its weak hypothesis.

Now we run the $\alpha$-Boost boosting algorithm [27, Section 6.4.2] on data set $S'$, but using the robust loss rather than 0-1 loss. That is, we start with $D_1$ uniform on $S'$.[4] Then for each round $t$, we get $h_{D_t}$ as a weak robust classifier with respect to $D_t$, and for each $(x, y) \in S'$ we define a distribution $D_{t+1}$ over $S'$ satisfying

$$D_{t+1}(\{(x, y)\}) \propto D_t(\{(x, y)\}) \exp\{-2\alpha \mathbb{1}[\forall x' \in \mathcal{U}(x), h_{D_t}(x') = y]\},$$

where $\alpha$ is a parameter we can set. Following the argument from [27, Section 6.4.2], after $T$ rounds we are guaranteed

$$\min_{(x,y) \in S'} \frac{1}{T} \sum_{t=1}^{T} \mathbb{1}[\forall x' \in \mathcal{U}(x), h_{D_t}(x') = y] \geq \frac{2}{3} - \frac{2}{3}\alpha - \frac{\ln(|S'|)}{2\alpha T},$$

so we will plan on running until round $T = 1 + 48 \ln(|S'|)$ with value $\alpha = 1/8$ to guarantee

$$\min_{(x,y) \in S'} \frac{1}{T} \sum_{t=1}^{T} \mathbb{1}[\forall x' \in \mathcal{U}(x), h_{D_t}(x') = y] > \frac{1}{2},$$

so that the classifier $\hat{h}(x) := \mathbb{1}\left[\frac{1}{T} \sum_{t=1}^{T} h_{D_t}(x) \geq \frac{1}{2}\right]$ has $\hat{R}_{\mathcal{U}}(\hat{h}; S') = 0$.

Furthermore, note that, since each $h_{D_t}$ is given by $\mathbb{A}(S_{D_t})$, where $S_{D_t}$ is an $m_0$-tuple of points in $S'$, the classifier $\hat{h}$ is specified by an ordered sequence of $m_0 T$ points from $S$. Altogether, $\hat{h}$ is a function specified by an ordered sequence of $m_0 T$ points from $S$, and which has

$$\hat{R}_{\mathcal{U}}(\hat{h}; S) \leq \min_{h \in \mathcal{H}} \hat{R}_{\mathcal{U}}(h; S).$$

Similarly to the realizable case (see the proof of Lemma 13), uniform convergence guarantees for sample compression schemes [see 15] remain valid for the robust loss, by essentially the same argument; the essential argument is the same as in the proof of Lemma 13 except using Hoeffding's inequality to get concentration of the empirical robust risks for each fixed index sequence, and then a union bound over the possible index sequences as before. We omit the details for brevity. In particular, denoting $T_m = 1 + 48 \ln(m)$, for $m > m_0 T_m$, with probability at least $1 - \delta/2$,

$$R_{\mathcal{U}}(\hat{h}; \mathcal{D}) \leq \hat{R}_{\mathcal{U}}(\hat{h}; S) + \sqrt{\frac{m_0 T_m \ln(m) + \ln(2/\delta)}{2m - 2m_0 T_m}}.$$

Let $h^* = \operatorname{argmin}_{h \in \mathcal{H}} R_{\mathcal{U}}(h; \mathcal{D})$ (supposing the min is realized, for simplicity; else we could take an $h^*$ with very-nearly minimal risk). By Hoeffding's inequality, with probability at least $1 - \delta/2$,

$$\hat{R}_{\mathcal{U}}(h^*; S) \leq R_{\mathcal{U}}(h^*; \mathcal{D}) + \sqrt{\frac{\ln(2/\delta)}{2m}}.$$

By the union bound, if $m \geq 2m_0 T_m$, with probability at least $1 - \delta$,

$$R_{\mathcal{U}}(\hat{h}; \mathcal{D}) \leq \min_{h \in \mathcal{H}} \hat{R}_{\mathcal{U}}(h; S) + \sqrt{\frac{m_0 T_m \ln(m) + \ln(2/\delta)}{m}}$$

$$\leq \hat{R}_{\mathcal{U}}(h^*; S) + \sqrt{\frac{m_0 T_m \ln(m) + \ln(2/\delta)}{m}}$$

$$\leq R_{\mathcal{U}}(h^*; \mathcal{D}) + 2\sqrt{\frac{m_0 T_m \ln(m) + \ln(2/\delta)}{m}}.$$

---

[4]We ignore the possibility of repeats; for our purposes we can just remove any repeats from $S'$ before this boosting step.

Since $T_m = O(\log(m))$, the above is at most $\varepsilon$ for an appropriate choice of sample size $m = O\left(\frac{m_0}{\varepsilon^2} \log^2\left(\frac{m_0}{\varepsilon}\right) + \frac{1}{\varepsilon^2} \log\left(\frac{1}{\delta}\right)\right)$. This concludes the upper bound on $\mathcal{M}^{\mathrm{ag}}_{\varepsilon,\delta}(\mathbb{B}; \mathcal{H}, \mathcal{U})$, and the lower bound trivially holds from the definition of $\mathbb{B}$. $\qquad\square$

## D  Finite Character Property

[4] gave a *formal* definition of the notion of "dimension" or "complexity measure", that all previously proposed dimensions in statistical learning theory comply with. In addition to characterizing learnability, a dimension should satisfy the *finite character* property:

**Definition 4** (Finite Character). A dimension characterizing learnability can be abstracted as a function $F$ that maps a class $\mathcal{H}$ to $\mathbb{N} \cup \{\infty\}$ and satisfies the *finite character* property: For every $d \in \mathbb{N}$ and $\mathcal{H}$, the statement "$F(\mathcal{H}) \geq d$" can be demonstrated by a finite set $X \subseteq \mathcal{X}$ of domain points, and a finite set of hypotheses $H \subseteq \mathcal{H}$. That is, "$F(\mathcal{H}) \geq d$" is equivalent to the existence of a bounded first order formula $\phi(\mathcal{X}, \mathcal{H})$ in which all the quantifiers are of the form: $\exists x \in \mathcal{X}, \forall x \in \mathcal{X}$ or $\exists h \in \mathcal{H}, \forall h \in \mathcal{H}$.

For example, the property "$\mathrm{vc}(\mathcal{H}) \geq d$" is a finite character property since it can be verified with a finite set of points $x_1 \ldots, x_d \in \mathcal{X}$ and a finite set of classifiers $h_1, \ldots, h_{2^d} \in \mathcal{H}$ that shatter these points, and a predicate $E(x, h) \equiv x \in h$ (i.e., the value $h(x)$). In our case, in addition to having a domain $\mathcal{X}$ and a hypothesis class $\mathcal{H}$, we also have a relation $\mathcal{U}$. In Claim 14, we argue that our dimension $\mathfrak{D}_{\mathcal{U}}(\mathcal{H})$ satisfies Definition 4, though unlike VC dimension, we do need $\forall$ quantifiers. Furthermore, we provably *cannot* verify the statement $\mathfrak{D}_{\mathcal{U}}(\mathcal{H}) \geq d$ by evaluating the predicate $E(x, h)$ on finitely many $x$'s and $h$'s, but we can verify it using a predicate $P_{\mathcal{U}}(x, h) \equiv \forall z \in \mathcal{U}(x) : h(z) = h(x)$ that evaluates the *robust* behavior of $h$ on $x$ w.r.t. $\mathcal{U}$. The proof is deferred to Appendix D.

**Claim 14.** $\mathfrak{D}_{\mathcal{U}}(\mathcal{H})$ *satisfies the finite character property of Definition 4.*

*Proof.* By the definition of $\mathfrak{D}_{\mathcal{U}}(\mathcal{H})$ in Equation 12, to demonstrate that $\mathfrak{D}_{\mathcal{U}}(\mathcal{H}) \geq d$, it suffices to present a finite subgraph $G = (V, E)$ of $G^{\mathcal{U}}_{\mathcal{H}} = (V_d, E_d)$ where every orientation $\mathcal{O} : E \to V$ has adversarial out-degree at least $\frac{n}{3}$. Since $V$ is, by definition, a finite collection of datasets robustly realizable with respect to $(\mathcal{H}, \mathcal{U})$ this means that we can demonstrate that $\mathfrak{D}_{\mathcal{U}}(\mathcal{H}) \geq d$ with a finite set $X \subseteq \mathcal{X}$ and a finite set of hypotheses $H \subseteq \mathcal{H}$ that can construct the finite collection $V$.

Note that in our case, we do not only have $\mathcal{X}$ and $\mathcal{H}$, but also a set relation $\mathcal{U}$ that specifies for each $x \in \mathcal{X}$ its corresponding set of perturbations $\mathcal{U}(x)$. We can still express $\mathfrak{D}_{\mathcal{U}}(\mathcal{H}) \geq d$ with a bounded formula using only quantifiers over $\mathcal{H}$ and $\mathcal{X}$, though unlike in the case of VC dimension, we do also need $\forall$ quantifiers. Furthermore, we provably cannot verify the formula by evaluating $h(x)$ only on finitely many $x \in \mathcal{X}, h \in \mathcal{H}$, since $\mathcal{U}(x)$ can be *infinite*. But, we can verify it given access to a predicate $P_{\mathcal{U}}(h, x) \equiv \forall z \in \mathcal{U}(x) : h(z) = h(x)$. $\qquad\square$