# OpenReview forum: "Adversarially Robust Learning: A Generic Minimax Optimal Learner and Characterization"
_NeurIPS.cc/2022/Conference — NeurIPS 2022 Accept_

### Official Review · Reviewer_VT2M · 2022-07-09

**Rating:** 7
**Confidence:** 4
**Soundness:** 4 excellent
**Presentation:** 4 excellent
**Contribution:** 4 excellent

**Summary:**

This paper studies the problem of learning predictors robust to adversarial examples presented at test-time. The main goal of the paper is to characterize adversarially robust learning for binary hypothesis classes and design an optimal learning algorithm for this task.

The main contributions of the paper are on three axes: first, it is proved that local learners (which were proposed in previous work) are not optimal; second, a novel global learner is provided, which is based on the well-known one-inclusion graph algorithm; this algorithm yields a generic minimax optimal learner for the problem of interest; third, based on the structure of this algorithm, a tight complexity measure is given by the authors. This quantity characterizes adversarially robust learnability in both the realizable and the agnostic settings. Finally, various conjectures are proposed for future work.

**Questions:**

I find the observation between the lines 307-311 very interesting. I would appreciate if the authors further comment on this point in greater detail.



**Limitations:**

The authors have extensively discussed future directions in order to extend their results.

**Strengths And Weaknesses:**

The paper is very well-written and clearly structured. The results are clear and very interesting. The structure of the one-inclusion algorithm has found various applications recently (such as the characterization of multiclass learnability and partial concept classes) and it is appealing that this is also the case in adversarially robust learning. The idea of the global one-inclusion graph algorithm is novel to the best of my knowledge and the intuition provided is clear. The only point that I do not find completely satisfying is the provided complexity measure, which is not as intuitive as other dimensions in the literature; however, I believe that the proposed dimension is the starting point of a research direction aiming at simplifying the characterization of  adversarially robust learnability.

In general, I believe that the paper is a strong contribution to the area of statistical learning theory.

---

> ### Author Response · Authors · 2022-08-02
> **Response to Reviewer VT2M**
>
> We thank you for your detailed review and valuable feedback!
>
> > I find the observation between the lines 307-311 very interesting. I would appreciate if the authors further comment on this point in greater detail.
>
> The robust shattering dimension (Definition 2) gives rise to a "cube" in our global one-inclusion graph, meaning it is a subgraph isomorphic to the Boolean cube {0,1}$^n$.  Specifically, any $z_1, \dots, z_k$ that are robustly shattered (as in Definition 2) can be used to construct a finite subgraph where the vertices are {$(x_1^{y_1},y_1), \dots, (x_k^{y_k},y_k)$} for $y \in ${$+1,-1$}.  By our definition of degree, these vertices will have full-degree, since every vertex will have a neighbour (with only a single label flipped).  In contrast, the full-degree dimension that we propose (in line 292)  gives *any* finite subgraph where every vertex has full-degree, including those not isomorphic to the Boolean cube.  Following the terminology in multiclass learning, we call this a "pseudo-cube", as it needn't be isomorphic to the Boolean cube.

---

### Official Review · Reviewer_pb28 · 2022-07-11

**Rating:** 8
**Confidence:** 4
**Soundness:** 4 excellent
**Presentation:** 4 excellent
**Contribution:** 4 excellent

**Summary:**

The authors give the first combinatorial characterization of adversarially robust PAC-learning (test-time adversarial perturbation), resolving an open problem of Montasser, Hanneke, and Srebro (2019) and closing an infinite gap in sample complexity seen by previous methods. The authors also provide a universal near-optimal algorithm based on a robust variant of the classical one-inclusion graph approach. In particular, the authors define an infinite object called the global inclusion graph G_n(H) roughly defined as follows:

$G_n(H)$ has vertex set given by all robustly realizable multisets of labeled examples $(x_1,y_1), \ldots, (x_n,y_n)$, and two vertices $z=(x,y)$,$z'=(x',y')$ have an edge if |z \Delta z'| =1, $y \neq y'$, and $U(x) \cap U(x')$ is nonempty (in fact an edge is defined for each element in this intersection). Given an orientation of edges in $G_n(H)$, the "out-degree" of a vertex $z=(x_1,y_1), \ldots, (x_n,y_n)$ is given by the number of indices that witness an edge.

The authors main result is to show that if there is an orientation of edges in $G_n(H)$ with small out-degree, then there exists a good robust learner. Moreover, if any good robust learner implies the existence of an orientation with small out-degree.

Note that this itself does not give a combinatorial characterization in the traditional sense (due to the global inclusion graph being infinite). To circumvent this issue, the authors introduce a finite variant still capturing the out-degree structure, and show that this parameter exactly characterizes robust learnability (up to log factors). Finally, it is worth noting that both the algorithm and dimension given are "global" in the sense that they must see the perturbation sets of elements outside their training set. The authors show this is in fact necessary, there exist robustly learnable classes that cannot be learned by any `local' algorithm that is restricted to information about its training set.



**Questions:**

I have no major questions, but a few minor comments:

1. It might be less confusing to change the name of your notion of degree rather than abusing notation.

2. DS-dimension should be mentioned at the beginning of section 5 re: characterizing multiclass learning

3. typo: integere

**Limitations:**

Yes

**Strengths And Weaknesses:**

This work resolves an important open problem in the study of adversarially robust learning, and provides very interesting insight on the power of local vs global robust algorithms. Many of the techniques build on known analysis of the one-inclusion graph and seem inspired by recent work on partial and multi-class learning, but the generalization to robust learning is certainly a non-trivial contribution. Overall, the paper is well-written and is likely to have high impact in the (theoretical) study of adversarially robust learning.

The main weakness of this work is that the proposed combinatorial dimension is quite complicated, lacking the simplicity of other combinatorial notions like VC-dimension (or even say DS-dimension) that often help lend these notions their broad impact across the field. That said, the authors do a very nice job discussing this issue and give a number of relevant conjectures in this direction.

---

> ### Author Response · Authors · 2022-08-02
> **Response to Reviewer pb28**
>
> We thank you for your detailed review and valuable feedback!
> We will fix the typo, update the notation/naming convention for our notion of degree to something less confusing, and mention the DS-dimension explicitly in the beginning of Section 5.

---

### Official Review · Reviewer_bV2J · 2022-07-11

**Rating:** 9
**Confidence:** 4
**Soundness:** 4 excellent
**Presentation:** 4 excellent
**Contribution:** 4 excellent

**Summary:**

The following points summarize the paper:
- The paper starts with showing that there are problem instances $(\mathcal{H}, \mathcal{U})$ which are not learnable by local lernears (with access to $\mathcal{U}(\cdot)$ on only training points). Using construction from later sections, they show that global learners can learn such problem instances.
- They introduce a novel global one-inclusion graph construction which uses $\mathcal{H}$ and $\mathcal{U}$ in a global manner.
- Using global one-inclusion graph, they also derive an optimal learner by orienting the edges of this graph to minimize a specific notion of out-degree.
-  They also propose a new dimension $\mathfrak{D}_\mathcal{U}$ and show that a problem instance $(\mathcal{H}, \mathcal{U})$ is robustly PAC-learnable iff $\mathfrak{D}_\mathcal{U}$ is finite. In particular, they provide tight (upto logarithmic factor) lower and upper bound of sample complexity based on $\mathfrak{D}_\mathcal{U}$. This closes potentially infinite gap of [21].
- They provide some ways to compute $\mathfrak{D}_\mathcal{U}$ and extend their results to agnostc case.

**Questions:**

Please see the previous section.

**Limitations:**

The authors have adequately addressed the limitations of their work.

**Strengths And Weaknesses:**

It is a strong theoretical paper with many novel theoretical contributions. The paper not only establishes that it is impossible for local learners to learn some of the problem instances robustly, it also provides a dimesnion measure to characterize robust learnability. The results are well presented and contrasted well with the existing results in the literature.

I did not notice any weaknesses. I have only one question: how easy/difficult is it to estimate $\mathfrak{D}_\mathcal{U}$ for a general case?

---

> ### Author Response · Authors · 2022-08-02
> **Response to Reviewer bV2J**
>
> We thank you for your detailed review and valuable feedback!
>
> > I did not notice any weaknesses. I have only one question: how easy/difficult is it to estimate $\mathfrak{D}_{\mathcal{U}}$ for a general case?
>
> Section 5.2 provides some examples calculating  $\mathfrak{D}_{\mathcal{U}}$,
>
> along with some basic upper bounds. But in general, we think that it is difficult to estimate $\mathfrak{D}_{\mathcal{U}}$
>
> for an arbitrary $(\mathcal{H}, \mathcal{U})$.
> To this end, we discuss specific conjectures in Section 5.3 (lines 285-315) that outline directions towards obtaining a simpler characterization (Conjecture 1) and obtaining tighter relations between $\mathfrak{D}_{\mathcal{U}}$ and the VC dimension ${\rm vc}(\mathcal{H})$ (Conjecture 3) than what we currently know (Proposition 8, line 263).

---

### Meta-Review · Area_Chair_LFHw · 2022-08-27

**Recommendation:** Accept
**Confidence:** Certain

**Metareview:**

A strong theoretical paper that presents novel results on adversarially robust learning algorithms. The paper designs minimax optimal robust learners and in the process identifies a key property namely locality, of existing algorithms that leads to sub-optimality. The paper also resolves an open question from Montasser et al.'19 by identifying a combinatorial quantity that is closely related to robust learning.

All the reviewers agree that this is a strong theory paper and should be accepted to NeurIPS.

**Award:**

No

---

### Decision · Program_Chairs · 2022-09-14

Accept